# Patch-wise Structural Loss for Time Series Forecasting

**Dilfira Kudrat** [1]   **Zongxia Xie** [1]   **Yanru Sun** [1]   **Tianyu Jia** [1]   **Qinghua Hu** [1]

## Abstract

Time-series forecasting has gained significant attention in machine learning due to its crucial role in various domains. However, most existing forecasting models rely heavily on point-wise loss functions like Mean Squared Error, which treat each time step independently and neglect the structural dependencies inherent in time series data, making it challenging to capture complex temporal patterns accurately. To address these challenges, we propose a novel **P**atch-wise **S**tructural (**PS**) loss, designed to enhance structural alignment by comparing time series at the patch level. Through leveraging local statistical properties, such as correlation, variance, and mean, PS loss captures nuanced structural discrepancies overlooked by traditional point-wise losses. Furthermore, it integrates seamlessly with point-wise loss, simultaneously addressing local structural inconsistencies and individual time-step errors. PS loss establishes a novel benchmark for accurately modeling complex time series data and provides a new perspective on time series loss function design. Extensive experiments demonstrate that PS loss significantly improves the performance of state-of-the-art models across diverse real-world datasets. The data and code are publicly available at: https://github.com/Dilfiraa/PS_Loss.

## 1. Introduction

Time series forecasting plays a crucial role across various real-world domains, including traffic (Kong et al., 2024; Long et al., 2024), weather (Lam et al., 2023; Wu et al., 2023b), and finance (Huang et al., 2024a), where accurate forecasting is essential for informed decision-making. Recent advancements in deep learning have developed models

[1]College of Intelligence and Computing, Tianjin University, China. Correspondence to: Zongxia Xie <caddiexie@hotmail.com>.

*Proceedings of the 42nd International Conference on Machine Learning*, Vancouver, Canada. PMLR 267, 2025. Copyright 2025 by the author(s).

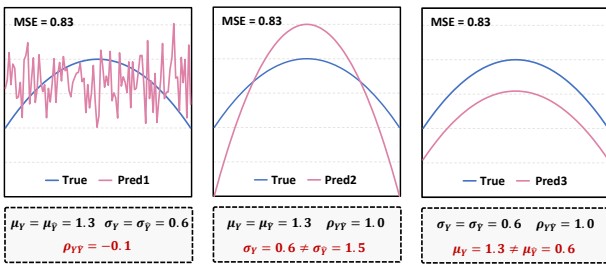

*Figure 1.* Limitations of MSE loss in capturing structural differences. Predictions with similar MSE but different statistical properties: (a) Poor linear correlation leads to misalignment in directionality and pattern; (b) Variance mismatch distorts the fluctuation dynamics; (c) Mean offset causes a shift in the overall level of the forecast.

that capture complex dependencies and intricate patterns in time series data, significantly improving forecasting accuracy (Zhou et al., 2021; 2022; Sun et al., 2024a; Yu et al., 2024b; Sun et al., 2024b; Yu et al., 2024a; Qiu et al., 2025b; Sun et al., 2025). These models aim to produce multi-step predictions based on historical data that closely follow the trajectory of future series.

Despite these advancements, most current forecasting models rely on Mean Squared Error (MSE) as the primary loss function. MSE evaluates predictions by averaging point-wise distances, which overlooks important structural characteristics inherent in time series data, leading to suboptimal forecasts that fail to accurately capture the underlying temporal dynamics (Le Guen & Thome, 2019; Lee et al., 2022; Cuturi & Blondel, 2017).

To illustrate the limitation of MSE, we consider three predictions with similar MSE values but different forecasting quality, as shown in Figure 1. The prediction in Figure 1a exhibits **poor correlation** with the ground truth, failing to capture the overall directionality and pattern. In contrast, the prediction in Figure 1b aligns with the general shape of the ground truth but fails to match its variability, as indicated by the **variance mismatch**, leading to distorted fluctuation dynamics. On the other hand, the prediction in Figure 1c maintains the shape but exhibits a **mean offset**, causing a consistent shift in the overall forecast level. These examples highlight that correlation, variance, and mean capture distinct yet complementary aspects of structural similarity,

providing a more comprehensive assessment of the alignment between prediction and ground truth.

However, measuring correlation, mean, and variance at a global level is insufficient, as global metrics often fail to capture local variations in statistical properties, especially in non-stationary time series where these properties evolve over time. This limitation highlights the need for a localized approach that can adapt to these variations and provide a more precise comparison of time series.

To address these challenges, we propose a **P**atch-wise **S**tructural (**PS**) loss that improves time series forecasting by introducing structural similarity metrics at the patch level. Our method comprises three key components. First, we introduce the Fourier-based Adaptive Patching (FAP) module, which adaptively segments the time series into patches, enabling a finer-grained assessment of structural alignment. Second, we propose the Patch-wise Structural (PS) loss, which integrates correlation, variance, and mean losses to capture local structural similarities and account for variations across different regions of the data. Third, we employ a Gradient-based Dynamic Weighting (GDW) strategy to adaptively balance the contributions of each loss term based on their gradients, ensuring that all structural aspects are adequately accounted for during training. PS loss can be seamlessly combined with MSE loss, resulting in more robust and accurate forecasting. Our contributions are as follows:

- We introduce a novel structural similarity measure that combines correlation, variance, and mean, offering a more comprehensive understanding of structural patterns and improving alignment between predictions and ground truth.

- We propose a patch-wise comparison method that enables localized similarity measures and accounts for changing statistical properties within the series, ensuring accurate structural alignment.

- Extensive experiments on five state-of-the-art architectures and seven real-world datasets confirm the effectiveness of our approach, demonstrating consistent performance gains and enhanced forecasting accuracy.

## 2. Related Work

### 2.1. Time Series Forecasting Models

In recent years, numerous deep forecasting models have been proposed to capture complex dependencies in time series data by leveraging the powerful representation capabilities of neural networks (Kim et al., 2025; Shao et al., 2024; Qiu et al., 2024; 2025a). Based on their architecture, these models can be categorized into CNN, Transformer, MLP, and LLM-based approaches. **CNN-based**

models excel at modeling local temporal relationships (Wu et al., 2023a; Luo & Wang, 2024; Liu et al., 2022). In contrast, **Transformer-based** models have gained significant attention for their ability to model long-range dependencies through self-attention mechanisms (Nie et al., 2023; Liu et al., 2024a; Chen et al., 2024b; Lin et al., 2024b; Zhou et al., 2022; Yu et al., 2023). Additionally, **MLP-based** models offer efficient alternatives that balance simplicity with performance (Zeng et al., 2023; Wang et al., 2024; Lin et al., 2024a; Ekambaram et al., 2023; Jia et al., 2025). Recently, **LLM-based** models have emerged as a promising direction, leveraging the capabilities of large language models to effectively capture complex temporal dependencies (Zhou et al., 2023; Liu et al., 2024b; Jin et al., 2023; Niu et al., 2025). These models use point-wise MSE as their loss function, treating each time step independently, which limits their ability to capture the structural nuances of time series data and may hinder forecasting performance.

### 2.2. Loss Functions for Time Series Forecasting

Recent efforts to address the limitations of MSE have given rise to alternative loss functions (Le Guen & Thome, 2019; Cuturi & Blondel, 2017; Lee et al., 2022; Wang et al., 2025), which can be broadly categorized into shape-focused losses and dependency-focused losses. **Shape-focused losses** aim to capture structural similarities between the ground truth and predictions by explicitly addressing shape misalignment. For instance, Dynamic Time Warping (DTW)-based methods such as Soft-DTW (Cuturi & Blondel, 2017) and DILATE (Le Guen & Thome, 2019) align sequences under temporal distortions. While effective at improving shape alignment, their high computational complexity limits scalability. TILDE-Q (Lee et al., 2022), on the other hand, introduces transformation invariance, making it robust to amplitude shifts, phase shifts, and scaling differences, thus focusing on shape similarity. **Dependency-focused losses** focus on capturing temporal dependencies within the forecast window. FreDF (Wang et al., 2025) bypasses label correlation complexity by learning to forecast in the frequency domain. Despite these advancements, most prior methods rely on global comparisons of entire time series, overlooking crucial localized structural details. In contrast, our proposed approach incorporates patch-wise statistical properties into the loss function, enabling a more granular structural measurement of the data and providing a fundamentally new perspective on time series loss design.

## 3. Methodology

### 3.1. Preliminaries

Given a historical time series $\mathbf{X} = \{x_0, x_1, \ldots, x_{L-1}\} \in \mathbb{R}^{C \times L}$, where $C$ is the number of channels and $L$ is the length of the lookback window. The goal of time series

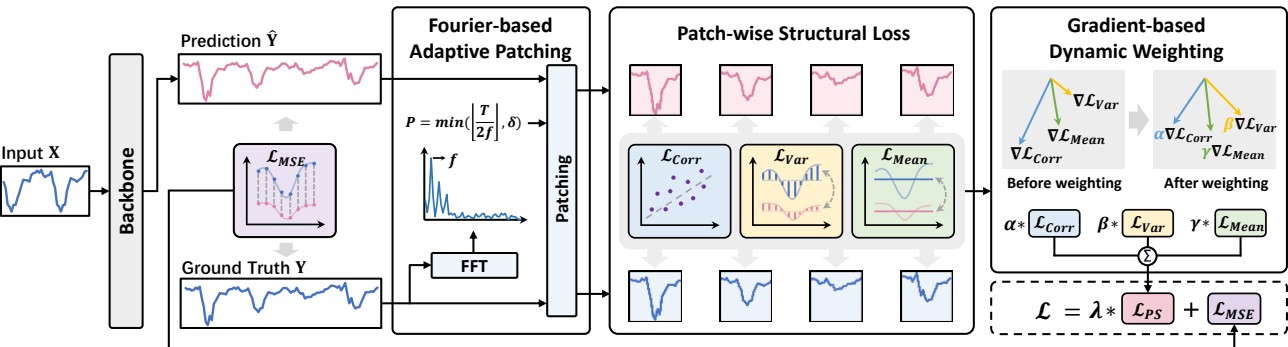

*Figure 2.* Overview of the proposed PS loss. The method consists of three main components: (1) Fourier-based Adaptive Patching, where the ground truth $Y$ and predicted series $\hat{Y}$ are adaptively segmented into patches; (2) Patch-wise Structural Loss, which measures local similarity between patches by integrating correlation ($\mathcal{L}_{Corr}$), variance ($\mathcal{L}_{Var}$), and mean losses ($\mathcal{L}_{Mean}$); and (3) Gradient-based Dynamic Weighting, which dynamically adjusts the weights of these loss components ($\alpha$, $\beta$, and $\gamma$) based on their gradient magnitudes to ensure balanced optimization. The PS loss ($\mathcal{L}_{PS}$) is seamlessly integrated to the MSE loss ($\mathcal{L}_{MSE}$) to improve forecasting accuracy.

forecasting is to learn a mapping $g : \mathbb{R}^{C \times L} \to \mathbb{R}^{C \times T}$ that generates predictions $\hat{\mathbf{Y}} = \{\hat{y}_0, \hat{y}_1, \ldots, \hat{y}_{T-1}\} \in \mathbb{R}^{C \times T}$ approximating the ground truth $\mathbf{Y} = \{y_0, y_1, \ldots, y_{T-1}\} \in \mathbb{R}^{C \times T}$, where $T$ is the forecasting length. For simplicity of notation, we focus on the univariate case ($C = 1$) in the following discussions. MSE is one of the most common loss functions for evaluating prediction accuracy, and it is defined as:

$$\mathcal{L}_{MSE} = \frac{1}{T} \sum_{i=0}^{T-1} (y_i - \hat{y}_i)^2. \tag{1}$$

### 3.2. Overview

Our method addresses the structural alignment limitations of MSE loss by focusing on localized pattern differences. To achieve this, we incorporate three key components: Fourier-based Adaptive Patching (FAP), Patch-wise Structural (PS) Loss, and Gradient-Based Dynamic Weighting (GDW), which collectively measure the patch-wise structural similarity between prediction and ground truth, as illustrated in Figure 2.

We first generate the prediction $\hat{\mathbf{Y}}$ using a backbone model, forming the basis for subsequent patch-level comparisons. Following this, **FAP** segments both the ground truth and prediction into patches based on the dominant frequency of the ground truth, ensuring that the patches capture recurring patterns and coherent structural information. Once the series is segmented, the **PS loss** evaluates the similarity between patches by integrating correlation, variance, and mean losses, enabling a comprehensive assessment of local structural alignment across the time series. To further refine the optimization process, **GDW** dynamically adjusts the weights of the three losses based on their gradients, ensuring balanced contributions from each component and leading to more effective optimization and consistent model training.

By seamlessly combining PS loss with MSE, we handle both global point-wise errors and local structural discrepancies. This integration enhances forecasting accuracy, demonstrating the advantages of incorporating patch-wise structural analysis into time series loss functions.

### 3.3. Fourier-based Adaptive Patching

To enable localized structural similarity measures between time series, we propose a patch-wise comparison approach that divides the prediction $\hat{\mathbf{Y}} \in \mathbb{R}^T$ and the ground truth $\mathbf{Y} \in \mathbb{R}^T$ into patches. By comparing these patches, we can better account for the structural patterns within the series.

In order to adaptively determine the patch length, we introduce the Fourier-based Adaptive Patching (FAP) technique. By analyzing the amplitude spectrum of the ground truth via the Fast Fourier Transform (FFT), we identify its dominant periodicity:

$$\mathbf{A} = \mathrm{Amp}\left(\mathcal{FFT}(\mathbf{Y})\right), \quad f = \operatorname*{arg\,max}_{f_* \in \{1, \ldots, \lfloor T/2 \rfloor\}} (\mathbf{A}), \tag{2}$$

where, $\mathrm{Amp}(\cdot)$ calculates the amplitude values and $f$ denotes the dominant frequency. The period $p = \lfloor T/f \rfloor$ serves as an initial patch length candidate, ensuring the patches reflect recurring structural patterns. However, dominant periodicity alone may produce overly large patches, missing finer structure nuances. To balance granularity and coherence, the final patch length is determined as:

$$P = \min\left(\left\lfloor \frac{p}{2} \right\rfloor, \delta\right), \quad p = \left\lfloor \frac{T}{f} \right\rfloor, \tag{3}$$

where $\delta$ is a predefined threshold. This adjustment ensures patches remain focused on localized structural patterns, even when the dominant periodicity is unclear or too large.

With patch length $P$ and stride $S = \lfloor P/2 \rfloor$, the series is segmented into patches $\hat{\mathbf{Y}}_p \in \mathbb{R}^{P \times N}$ and $\mathbf{Y}_p \in \mathbb{R}^{P \times N}$, where

$N = \lfloor (T - P)/S \rfloor + 1$ is the total number of patches. The $i$-th patch of $\mathbf{Y}$ is defined as $Y_p^{(i)} = \{y_0^{(i)}, y_1^{(i)}, \ldots, y_{P-1}^{(i)}\}$, where $y_j^{(i)}$ refers to the $y_{i \times S + j}$ in the unpatched time series.

### 3.4. Patch-wise Structural Loss

Building on the adaptive patching framework, we propose the Patch-wise Structural (PS) loss to enhance time series forecasting by capturing and aligning localized statistical properties within each patch. PS loss focuses on three key metrics, namely correlation, variance, and mean, to offer a deeper structural comparison.

**Correlation Loss.** Correlation quantifies the degree of directional and pattern consistency between the ground truth and predicted patches. We define the correlation loss based on the Pearson Correlation Coefficient (PCC) (Cohen et al., 2009), as follows:

$$
\begin{aligned}
\mathcal{L}_{Corr} &= \frac{1}{N} \sum_{i=0}^{N-1} 1 - \rho(Y_p^{(i)}, \hat{Y}_p^{(i)}) \\
&= \frac{1}{N} \sum_{i=0}^{N-1} 1 - \frac{\sum_{j=0}^{P-1}(y_j^{(i)} - \mu^{(i)})(\hat{y}_j^{(i)} - \hat{\mu}^{(i)})}{\sigma^{(i)} \hat{\sigma}^{(i)}},
\end{aligned}
\tag{4}
$$

where $\mu^{(i)}$ and $\sigma^{(i)}$ represent the mean and standard deviation of the $i$-th patch, respectively, and $\rho(\cdot, \cdot)$ denotes the PCC. Minimizing $\mathcal{L}_{Corr}$ encourages the model to align its predictions with the ground truth's directionality and trends. This alignment helps the model capture underlying pattern relationships, which are critical for time series forecasting tasks that require more than just point-wise accuracy.

**Variance Loss.** Variance quantifies the degree of fluctuation within a patch, providing a localized measure of variability. To ensure that predictions represent these localized dynamics, we introduce a variance loss that compares the relative dispersion between ground truth and predicted patches.

Rather than directly comparing variance values, our approach focuses on softly aligning the relative dispersion of each time step within patches. Specifically, we compute the deviations, $Y_p^{(i)} - \mu^{(i)}$ and $\hat{Y}_p^{(i)} - \hat{\mu}^{(i)}$, and normalize them into probability distributions using the softmax function. This transformation highlights the relative magnitude of the deviations, allowing us to focus on how the values within each patch are dispersed relative to one another (Shu et al., 2021). The similarity between these distributions is measured via the Kullback–Leibler (KL) divergence, leading to the following variance loss:

$$
\mathcal{L}_{Var} = \frac{1}{N} \sum_{i=0}^{N-1} \mathrm{KL}\left(\phi\left(Y_p^{(i)} - \mu^{(i)}\right) \| \phi\left(\hat{Y}_p^{(i)} - \hat{\mu}^{(i)}\right)\right),
\tag{5}
$$

where $\phi(\cdot)$ is the softmax function. Due to the shift-invariance property of softmax, i.e., $\phi(Y_p^{(i)} - \mu^{(i)}) = \phi(Y_p^{(i)})$, the variance loss formulation simplifies to:

$$
\mathcal{L}_{Var} = \frac{1}{N} \sum_{i=0}^{N-1} \mathrm{KL}\left(\phi\left(Y_p^{(i)}\right) \| \phi\left(\hat{Y}_p^{(i)}\right)\right).
\tag{6}
$$

By aligning the relative dispersion within patches, the variance loss encourages the model to capture the underlying fluctuation patterns of the time series, leading to more structurally coherent forecasts.

**Mean Loss.** The mean value of a patch provides a clear indicator of its central tendency. While correlation and variance losses effectively capture directionality patterns and localized variability, they may overlook systematic shifts or biases in the predictions. To address this, we introduce a mean loss term that directly measures the consistency of average values.

Specifically, the mean loss is defined as the Mean Absolute Error (MAE) between the mean values of the ground truth and predicted patches:

$$
\mathcal{L}_{Mean} = \frac{1}{N} \sum_{i=0}^{N-1} \left| \mu^{(i)} - \hat{\mu}^{(i)} \right|,
\tag{7}
$$

where $\mu^{(i)}$ and $\hat{\mu}^{(i)}$ represent the mean values of the $i$-th patch in the ground truth and predictions, respectively. By aligning the mean values, the mean loss corrects prediction biases and ensures the model maintains accurate overall levels and central tendencies, enabling it to adapt to dynamic shifts in the time series.

**PS Loss.** Finally, we integrate the correlation, variance, and mean losses into a unified Patch-wise Structural (PS) loss, which is formulated as the weighted sum of the three individual loss components:

$$
\mathcal{L}_{PS} = \alpha \mathcal{L}_{Corr} + \beta \mathcal{L}_{Var} + \gamma \mathcal{L}_{Mean},
\tag{8}
$$

where $\alpha$, $\beta$, and $\gamma$ balance the contributions of the individual loss components. This integration provides a comprehensive structural alignment by leveraging the complementary strengths of each loss.

### 3.5. Gradient-based Dynamic Weighting

PS loss combines three loss components to measure patch-level similarity. To ensure effective training, it is crucial to balance these components, preventing any single loss term from dominating the optimization process (Chen et al., 2018). In multi-task learning, a common approach to balancing multiple losses is to weight them based on their gradient magnitudes, which is robust to differences in loss

scales and directly reflects each loss's influence on the optimization process (Sener & Koltun, 2018; Lin et al., 2019; Liu et al.). Inspired by this, we propose a Gradient-based Dynamic Weighting (GDW) strategy, which adaptively adjusts loss weights based on gradients to achieve balanced optimization.

At training step $t$, let $W$ denote the model's output layer parameters. We compute the L2-norm of each loss component's gradient with respect to $W$:

$$G_{Corr}^{(t)} = \|\nabla_W \mathcal{L}_{Corr}^{(t)}\|_2, \quad G_{Var}^{(t)} = \|\nabla_W \mathcal{L}_{Var}^{(t)}\|_2,$$
$$G_{Mean}^{(t)} = \|\nabla_W \mathcal{L}_{Mean}^{(t)}\|_2, \tag{9}$$

where $G_{Corr}^{(t)}$ is the gradient magnitude of the correlation loss, and $G_{Var}^{(t)}$, $G_{Mean}^{(t)}$ are defined analogously. Using these gradient magnitudes, we compute the average gradient magnitude:

$$\overline{G}^{(t)} = \frac{G_{Corr}^{(t)} + G_{Var}^{(t)} + G_{Mean}^{(t)}}{3}. \tag{10}$$

The loss weights at iteration $t$ are then set as:

$$\alpha^{(t)} = \frac{\overline{G}^{(t)}}{G_{Corr}^{(t)}}, \quad \beta^{(t)} = \frac{\overline{G}^{(t)}}{G_{Var}^{(t)}}, \quad \gamma^{(t)} = c \cdot v \cdot \frac{\overline{G}^{(t)}}{G_{Mean}^{(t)}}. \tag{11}$$

To further refine the mean loss contribution, we introduce two scaling factors, $c$ and $v$, which adjust the weight of the mean loss based on the model's performance in correlation and variance. These factors are defined as:

$$c = \frac{1}{2} \cdot (1 + \frac{\sigma_{Y\hat{Y}}}{\sigma_Y \sigma_{\hat{Y}}}), \quad v = \frac{2\sigma_Y \sigma_{\hat{Y}}}{\sigma_Y^2 + \sigma_{\hat{Y}}^2}, \tag{12}$$

where $\sigma_{Y\hat{Y}}$ is the covariance between ground truth and predictions, and $\sigma_Y$ and $\sigma_{\hat{Y}}$ are their standard deviations. Both $c$ and $v$ lie in the range $[0, 1]$, increasing as the predictions align more closely with the ground truth in terms of correlation and variance. As alignment improves, the relative weight of the mean loss increases, ensuring it plays a greater role when correlation and variance are well-matched, thereby enhancing the refinement of predictions.

Finally, the total loss function for optimizing the time series forecasting model is formulated as the sum of the standard MSE loss and the PS loss:

$$\mathcal{L} = \mathcal{L}_{MSE} + \lambda \mathcal{L}_{PS}, \tag{13}$$

where $\lambda$ is a hyperparameter that controls the contribution of PS loss.

## 4. Experiments

### 4.1. Experimental Setup

**Datasets.** We conduct our experiments on seven real-world multivariate time series datasets, including ETT (ETTh1, ETTh2, ETTm1, ETTm2), Weather, ECL, and Exchange. For data preprocessing, we follow the standard protocol used by backbone studies. The detailed description of the datasets is provided in Appendix A.

**Backbones.** We select five state-of-the-art (SOTA) time series forecasting models with diverse architectures to comprehensively evaluate PS loss function. Specifically, we include Transformer-based models: iTransformer (Liu et al., 2024a), PatchTST (Nie et al., 2023); MLP-based models: DLinear (Zeng et al., 2023), TimeMixer (Wang et al., 2024); CNN-based model: TimesNet (Wu et al., 2023a).

**Implementation Details.** We use the official implementations of each backbone from their respective repositories for our experiments. For fair evaluation, when integrating PS loss to enhance the backbone model's performance, we follow their original experimental and hyperparameter settings, while only tuning the PS loss weight $\lambda$ and the patch length threshold $\delta$. The experiments are implemented using PyTorch and executed on NVIDIA RTX 3090 24GB GPU.

### 4.2. Main Results

We present the MSE and MAE on seven real-world datasets for five SOTA long-term multivariate forecasting models in Table 1. Notably, PS loss achieves a remarkable improvement on DLinear, with average reductions of 5.22% in MSE and 4.39% in MAE. Similar performance gains are observed across other backbone models and PS loss consistently enhances the forecasting performance significantly outperforming MSE loss in 134 out of 140 cases, further validating the robustness and broad applicability of the proposed loss function. These improvements highlight the unique strength of PS loss in capturing localized structural patterns within time series data.

### 4.3. Results on LLM-based Models

To further evaluate the generalizability and effectiveness of the proposed PS loss, we conduct experiments on two LLM-based time series forecasting models: OFA (Zhou et al., 2023), built on GPT-2 (Radford et al.) and Autotimes (Liu et al., 2024b), which utilizes LLaMA (Touvron et al., 2023).

The results in Table 2 confirm that incorporating PS loss produces consistent performance improvements over the standard MSE loss. These findings highlight the potential of PS loss to serve as a universal enhancement for LLM-based time series models, further solidifying its value as a powerful tool for advancing multivariate time series forecasting.

*Table 1.* Long-term multivariate forecasting results. The table reports MSE and MAE for different forecasting lengths $T \in \{96, 192, 336, 720\}$. The input sequence length is consistent with the backbone setting. The better results are highlighted in **bold**.

| Models | | iTransformer (2024a) | | | | PatchTST (2023) | | | | TimeMixer (2024) | | | | DLinear (2023) | | | | TimseNet (2023a) | | | |
|---|---|---|---|---|---|---|---|---|---|---|---|---|---|---|---|---|---|---|---|---|---|
| Loss Functions | | MSE | | + PS | | MSE | | + PS | | MSE | | + PS | | MSE | | + PS | | MSE | | + PS | |
| Metric | | MSE | MAE | MSE | MAE | MSE | MAE | MSE | MAE | MSE | MAE | MSE | MAE | MSE | MAE | MSE | MAE | MSE | MAE | MSE | MAE |
| ETTh1 | 96 | 0.387 | 0.405 | **0.379** | **0.396** | 0.375 | 0.400 | **0.368** | **0.394** | 0.385 | 0.402 | **0.366** | **0.392** | 0.384 | 0.405 | **0.367** | **0.389** | 0.384 | **0.405** | **0.382** | 0.406 |
| | 192 | 0.441 | 0.436 | **0.428** | **0.424** | 0.414 | 0.421 | **0.407** | **0.415** | 0.443 | 0.430 | **0.421** | **0.421** | 0.443 | 0.450 | **0.402** | **0.411** | 0.443 | 0.450 | **0.424** | **0.432** |
| | 336 | 0.491 | 0.462 | **0.474** | **0.453** | 0.432 | 0.436 | **0.420** | **0.426** | 0.512 | 0.470 | **0.489** | **0.453** | 0.447 | 0.448 | **0.435** | **0.435** | 0.494 | 0.471 | **0.457** | **0.448** |
| | 720 | 0.509 | 0.494 | **0.480** | **0.478** | 0.450 | 0.466 | **0.425** | **0.449** | 0.497 | 0.476 | **0.474** | **0.464** | 0.504 | 0.515 | **0.463** | **0.484** | 0.504 | 0.515 | **0.477** | **0.477** |
| | Avg | 0.457 | 0.449 | **0.440** | **0.438** | 0.418 | 0.431 | **0.405** | **0.421** | 0.460 | 0.444 | **0.437** | **0.432** | 0.445 | 0.454 | **0.417** | **0.430** | 0.457 | 0.460 | **0.435** | **0.441** |
| ETTh2 | 96 | 0.301 | 0.350 | **0.289** | **0.340** | 0.274 | 0.336 | **0.272** | **0.332** | 0.291 | 0.342 | **0.284** | **0.330** | 0.290 | 0.353 | **0.280** | **0.341** | 0.330 | 0.367 | **0.316** | **0.354** |
| | 192 | 0.380 | 0.399 | **0.373** | **0.391** | 0.339 | 0.380 | **0.336** | **0.375** | 0.376 | 0.396 | **0.355** | **0.381** | 0.388 | 0.422 | **0.358** | **0.396** | 0.405 | 0.415 | **0.387** | **0.396** |
| | 336 | 0.424 | 0.432 | **0.416** | **0.428** | 0.331 | 0.380 | **0.324** | **0.377** | 0.437 | 0.439 | **0.412** | **0.423** | 0.463 | 0.473 | **0.435** | **0.450** | 0.454 | 0.451 | **0.428** | **0.431** |
| | 720 | 0.430 | 0.447 | **0.421** | **0.441** | 0.378 | 0.421 | **0.376** | **0.417** | 0.464 | 0.464 | **0.423** | **0.440** | 0.733 | 0.606 | **0.597** | **0.540** | **0.434** | **0.448** | 0.438 | 0.448 |
| | Avg | 0.384 | 0.407 | **0.375** | **0.400** | 0.331 | 0.379 | **0.327** | **0.375** | 0.392 | 0.410 | **0.369** | **0.394** | 0.469 | 0.463 | **0.417** | **0.432** | 0.406 | 0.420 | **0.392** | **0.407** |
| ETTm1 | 96 | 0.342 | 0.377 | **0.326** | **0.360** | 0.288 | 0.342 | **0.284** | **0.334** | 0.328 | 0.364 | **0.316** | **0.350** | 0.301 | 0.345 | **0.296** | **0.339** | 0.334 | 0.375 | **0.329** | **0.369** |
| | 192 | 0.383 | 0.396 | **0.374** | **0.384** | 0.334 | 0.372 | **0.328** | **0.364** | 0.364 | 0.382 | **0.357** | **0.373** | 0.336 | 0.366 | **0.333** | **0.362** | 0.406 | 0.413 | **0.388** | **0.400** |
| | 336 | 0.418 | 0.418 | **0.410** | **0.406** | 0.367 | 0.393 | **0.357** | **0.383** | 0.387 | 0.402 | **0.385** | **0.394** | 0.372 | 0.389 | **0.365** | **0.380** | 0.415 | 0.422 | **0.394** | **0.406** |
| | 720 | 0.487 | 0.457 | **0.472** | **0.440** | 0.417 | 0.422 | **0.406** | **0.412** | 0.472 | 0.449 | **0.444** | **0.432** | 0.427 | 0.423 | **0.419** | **0.413** | 0.511 | 0.472 | **0.469** | **0.444** |
| | Avg | 0.408 | 0.412 | **0.396** | **0.397** | 0.352 | 0.382 | **0.344** | **0.373** | 0.388 | 0.399 | **0.375** | **0.387** | 0.359 | 0.381 | **0.353** | **0.374** | 0.417 | 0.421 | **0.395** | **0.405** |
| ETTm2 | 96 | 0.186 | 0.272 | **0.175** | **0.254** | 0.164 | 0.253 | **0.160** | **0.247** | 0.175 | 0.257 | **0.171** | **0.252** | 0.172 | 0.267 | **0.163** | **0.251** | 0.189 | 0.266 | **0.180** | **0.257** |
| | 192 | 0.254 | 0.314 | **0.242** | **0.299** | 0.221 | 0.291 | **0.216** | **0.287** | 0.240 | 0.302 | **0.234** | **0.295** | 0.237 | 0.314 | **0.222** | **0.296** | 0.263 | 0.312 | **0.246** | **0.298** |
| | 336 | 0.316 | 0.351 | **0.304** | **0.338** | 0.277 | 0.329 | **0.267** | **0.321** | 0.303 | 0.343 | **0.291** | **0.331** | 0.295 | 0.359 | **0.277** | **0.332** | 0.326 | 0.354 | **0.303** | **0.334** |
| | 720 | 0.414 | 0.407 | **0.401** | **0.394** | 0.365 | 0.384 | **0.357** | **0.378** | 0.396 | 0.400 | **0.386** | **0.389** | 0.427 | 0.439 | **0.377** | **0.397** | 0.418 | 0.405 | **0.418** | **0.402** |
| | Avg | 0.292 | 0.336 | **0.281** | **0.321** | 0.257 | 0.314 | **0.250** | **0.308** | 0.278 | 0.326 | **0.270** | **0.317** | 0.283 | 0.345 | **0.260** | **0.319** | 0.299 | 0.334 | **0.286** | **0.323** |
| Weather | 96 | 0.176 | 0.216 | **0.167** | **0.203** | 0.151 | 0.198 | **0.149** | **0.190** | 0.165 | 0.212 | **0.161** | **0.201** | 0.174 | 0.233 | **0.171** | **0.222** | 0.172 | 0.220 | **0.170** | **0.219** |
| | 192 | 0.227 | 0.260 | **0.219** | **0.249** | 0.196 | 0.242 | **0.193** | **0.235** | 0.211 | 0.254 | **0.207** | **0.244** | 0.218 | 0.278 | **0.212** | **0.259** | **0.225** | 0.264 | **0.225** | 0.263 |
| | 336 | 0.282 | 0.299 | **0.274** | **0.291** | 0.248 | 0.282 | **0.245** | **0.276** | 0.263 | 0.293 | **0.263** | **0.285** | 0.263 | 0.314 | **0.258** | **0.300** | **0.281** | 0.304 | **0.281** | 0.303 |
| | 720 | 0.357 | 0.348 | **0.353** | **0.343** | 0.319 | 0.335 | **0.318** | **0.330** | 0.343 | 0.345 | **0.340** | **0.337** | 0.332 | 0.374 | **0.323** | **0.356** | 0.359 | 0.354 | **0.358** | **0.352** |
| | Avg | 0.261 | 0.281 | **0.253** | **0.272** | 0.228 | 0.264 | **0.226** | **0.258** | 0.245 | 0.276 | **0.243** | **0.267** | 0.247 | 0.300 | **0.241** | **0.284** | 0.259 | 0.285 | **0.259** | **0.284** |
| ECL | 96 | 0.148 | 0.239 | **0.146** | **0.236** | 0.130 | 0.223 | **0.128** | **0.220** | 0.153 | 0.245 | **0.152** | **0.245** | 0.140 | 0.237 | **0.140** | **0.236** | 0.168 | 0.272 | **0.164** | **0.266** |
| | 192 | 0.167 | 0.258 | **0.161** | **0.250** | 0.149 | 0.240 | **0.145** | **0.235** | 0.166 | 0.257 | **0.165** | **0.256** | 0.154 | 0.250 | **0.153** | **0.248** | 0.186 | 0.289 | **0.177** | **0.277** |
| | 336 | 0.178 | 0.271 | **0.174** | **0.265** | 0.165 | 0.257 | **0.161** | **0.252** | 0.185 | 0.275 | **0.182** | **0.272** | 0.169 | 0.268 | **0.169** | **0.266** | 0.196 | 0.297 | **0.192** | **0.292** |
| | 720 | 0.211 | 0.300 | **0.208** | **0.297** | 0.208 | 0.296 | **0.196** | **0.282** | 0.224 | **0.312** | **0.223** | **0.312** | 0.204 | 0.300 | **0.203** | **0.297** | 0.235 | 0.329 | **0.220** | **0.314** |
| | Avg | 0.176 | 0.267 | **0.172** | **0.262** | 0.163 | 0.254 | **0.158** | **0.247** | 0.182 | 0.272 | **0.180** | **0.271** | 0.167 | 0.264 | **0.166** | **0.262** | 0.196 | 0.297 | **0.188** | **0.287** |
| Exchange | 96 | **0.086** | **0.206** | 0.087 | 0.206 | 0.093 | 0.214 | **0.089** | **0.208** | 0.086 | 0.204 | **0.084** | **0.202** | 0.085 | 0.209 | **0.078** | **0.199** | **0.105** | **0.235** | 0.107 | 0.236 |
| | 192 | 0.181 | 0.304 | **0.180** | **0.303** | 0.194 | 0.315 | **0.186** | **0.307** | 0.187 | 0.306 | **0.186** | **0.306** | 0.162 | 0.296 | **0.156** | **0.290** | 0.232 | 0.351 | **0.215** | **0.337** |
| | 336 | 0.338 | 0.422 | **0.336** | **0.420** | 0.355 | 0.436 | **0.345** | **0.427** | 0.386 | 0.454 | **0.332** | **0.417** | 0.333 | 0.441 | **0.325** | **0.432** | 0.393 | 0.462 | **0.377** | **0.450** |
| | 720 | **0.853** | **0.696** | 0.860 | 0.700 | **0.903** | **0.712** | 0.904 | 0.713 | 0.928 | 0.727 | **0.917** | **0.719** | 0.898 | 0.725 | **0.811** | **0.690** | 1.011 | 0.768 | **1.010** | 0.768 |
| | Avg | **0.364** | **0.407** | 0.366 | 0.407 | 0.386 | 0.419 | **0.381** | **0.414** | 0.397 | 0.423 | **0.380** | **0.411** | 0.381 | 0.420 | **0.373** | **0.412** | 0.435 | 0.454 | **0.427** | **0.448** |

*Table 2.* LLM-based long-term multivariate forecasting results. The table reports MSE and MAE for different forecasting lengths $T \in \{96, 192, 336, 720\}$. The input sequence length is consistent with the backbone setting. The better results are highlighted in **bold**.

| Models | Loss Functions | Dataset | ETTh1 | | | | ETTh2 | | | | ETTm1 | | | | ETTm2 | | | |
|---|---|---|---|---|---|---|---|---|---|---|---|---|---|---|---|---|---|---|
| | | Pred Len | 96 | 192 | 336 | 720 | 96 | 192 | 336 | 720 | 96 | 192 | 336 | 720 | 96 | 192 | 336 | 720 |
| OFA (2023) | MSE | MSE | 0.382 | 0.424 | 0.444 | 0.445 | 0.292 | 0.359 | 0.379 | 0.418 | 0.298 | 0.336 | 0.368 | 0.419 | 0.170 | 0.230 | 0.285 | 0.364 |
| | | MAE | 0.402 | 0.425 | 0.437 | 0.458 | 0.348 | 0.398 | 0.414 | 0.454 | 0.344 | 0.375 | 0.392 | 0.425 | 0.263 | 0.306 | 0.345 | 0.392 |
| | + PS | MSE | **0.371** | **0.413** | **0.434** | **0.422** | **0.283** | **0.355** | **0.372** | **0.398** | **0.290** | **0.331** | **0.364** | **0.416** | **0.162** | **0.218** | **0.275** | **0.355** |
| | | MAE | **0.393** | **0.416** | **0.430** | **0.445** | **0.338** | **0.381** | **0.402** | **0.431** | **0.343** | **0.367** | **0.388** | **0.418** | **0.248** | **0.289** | **0.329** | **0.375** |
| AutoTmies (2024b) | MSE | MSE | 0.360 | 0.388 | 0.401 | 0.405 | 0.295 | 0.359 | 0.392 | 0.465 | 0.289 | 0.337 | 0.373 | 0.430 | 0.181 | 0.244 | 0.299 | 0.372 |
| | | MAE | 0.400 | 0.419 | 0.429 | 0.440 | 0.355 | 0.400 | 0.431 | 0.483 | 0.345 | 0.374 | 0.396 | 0.432 | 0.269 | 0.311 | 0.349 | 0.398 |
| | + PS | MSE | **0.358** | **0.386** | **0.398** | **0.404** | **0.287** | **0.348** | **0.370** | **0.422** | **0.280** | **0.327** | **0.360** | **0.416** | **0.171** | **0.231** | **0.284** | **0.366** |
| | | MAE | **0.398** | **0.415** | **0.425** | **0.438** | **0.347** | **0.387** | **0.410** | **0.450** | **0.337** | **0.366** | **0.387** | **0.421** | **0.258** | **0.299** | **0.335** | **0.387** |

## 4.4. Comparison with Other Loss Functions

We evaluate the proposed PS loss against advanced loss functions. TILDE-Q (Lee et al., 2022) emphasizes shape similarity using transformation-invariant loss terms, while FreDF (Wang et al., 2025) bypasses label correlation by comparing time series in the frequency domain. The results are shown in Table 3. PS loss achieves the lowest MSE and MAE in most cases across various datasets and forecasting horizons. This is due to its ability to effectively measure localized structural similarity by evaluating correlation, variance, and mean at the patch level, which enables a more precise alignment between the ground truth and predictions.

## 4.5. Ablation Study

To evaluate the contribution of each component within the proposed PS loss, we conducted ablation experiments on the ETTh1 and weather datasets using DLinear as the backbone models. The results are presented in Table 4, with additional

*Table 3.* Comparison between the proposed Patch-wise Structural loss and other loss functions. The model is iTransformer and we report the result of three datasets-ETTh1, ETTm1, and Weather. The best results are highlighted in **bold**, and the second-best results are highlighted in underline.

| Dataset | | ETTh1 | | | | ETTm1 | | | | Weather | | | |
|---|---|---|---|---|---|---|---|---|---|---|---|---|---|
| Forecast length | | 96 | 192 | 336 | 720 | 96 | 192 | 336 | 720 | 96 | 192 | 336 | 720 |
| MSE | MSE | 0.387 | 0.441 | 0.491 | 0.509 | 0.342 | 0.383 | 0.418 | 0.487 | 0.176 | 0.227 | 0.282 | 0.357 |
| | MAE | 0.405 | 0.436 | 0.462 | 0.494 | 0.377 | 0.396 | 0.418 | 0.457 | 0.216 | 0.260 | 0.299 | 0.348 |
| TILDE-Q | MSE | 0.386 | 0.434 | 0.477 | 0.521 | 0.336 | 0.382 | 0.418 | 0.483 | 0.172 | 0.221 | 0.279 | 0.356 |
| (2022) | MAE | 0.401 | 0.429 | 0.454 | 0.502 | 0.366 | 0.392 | 0.416 | 0.453 | 0.208 | 0.252 | 0.296 | 0.346 |
| FreDF | MSE | 0.381 | 0.433 | **0.471** | 0.487 | 0.334 | 0.380 | 0.416 | 0.486 | 0.170 | 0.222 | 0.279 | 0.357 |
| (2025) | MAE | 0.396 | 0.426 | **0.446** | 0.481 | 0.367 | 0.393 | 0.414 | 0.451 | 0.210 | 0.254 | 0.297 | 0.348 |
| **+ PS** | MSE | **0.379** | **0.428** | 0.474 | **0.480** | **0.326** | **0.374** | **0.410** | **0.472** | **0.167** | **0.219** | **0.274** | **0.353** |
| | MAE | **0.396** | **0.424** | 0.453 | **0.478** | **0.360** | **0.384** | **0.406** | **0.440** | **0.203** | **0.249** | **0.291** | **0.343** |

*Table 4.* Ablation study of the components of PS loss on the ETTh1 and Weather datasets using DLinear as a backbone.

| Method | | Dlinear + PS | | w/o $\mathcal{L}_{Corr}$ | | w/o $\mathcal{L}_{Var}$ | | w/o $\mathcal{L}_{Mean}$ | | w/o Patching | | w/o Weighting | |
|---|---|---|---|---|---|---|---|---|---|---|---|---|---|
| Metric | | MSE | MAE | MSE | MAE | MSE | MAE | MSE | MAE | MSE | MAE | MSE | MAE |
| ETTh1 | 96 | **0.367** | **0.389** | 0.381 | 0.402 | 0.380 | 0.399 | 0.372 | 0.395 | 0.380 | 0.401 | 0.384 | 0.404 |
| | 192 | **0.402** | **0.411** | 0.439 | 0.446 | 0.440 | 0.444 | 0.404 | 0.414 | 0.436 | 0.441 | 0.403 | 0.411 |
| | 336 | **0.435** | **0.435** | 0.443 | 0.445 | 0.439 | 0.486 | 0.440 | 0.443 | 0.442 | 0.442 | 0.439 | 0.440 |
| | 720 | **0.463** | **0.484** | 0.510 | 0.519 | 0.509 | 0.517 | 0.474 | 0.493 | 0.522 | 0.528 | 0.528 | 0.532 |
| Weather | 96 | **0.171** | **0.222** | 0.174 | 0.233 | 0.172 | 0.223 | 0.172 | 0.223 | 0.171 | 0.224 | 0.172 | 0.226 |
| | 192 | **0.212** | **0.259** | 0.222 | 0.286 | 0.217 | 0.271 | 0.217 | 0.271 | 0.220 | 0.280 | 0.214 | 0.268 |
| | 336 | **0.258** | **0.300** | 0.261 | 0.311 | 0.259 | 0.303 | 0.259 | 0.303 | 0.263 | 0.314 | 0.261 | 0.307 |
| | 720 | **0.323** | **0.356** | 0.329 | 0.369 | 0.328 | 0.365 | 0.328 | 0.365 | 0.331 | 0.372 | 0.332 | 0.370 |

results provided in Appendix I.

**Loss components.** Removing any one of the three loss components leads to noticeable performance degradation across all horizons. This highlights the synergistic relationship between these components in capturing localized structural similarities.

**Adaptive patching.** To evaluate the importance of patching, we conducted a direct comparison by processing the entire time series without segmentation. This approach significantly worsens performance, especially for longer horizons, underscoring the value of the patching mechanism in capturing fine-grained patterns.

**Dynamic weighting.** To evaluate the importance of dynamic weighting, we replaced the adaptive weighting with a fixed weight of 1.0 for all three loss terms. This static weighting strategy results in performance degradation, demonstrating that dynamic weighting effectively balances each loss term and improves optimization and forecasting.

### 4.6. Zero-shot Forecasting

To evaluate how PS loss improves generalization to unseen datasets, we conducted zero-shot forecasting experiments. Following prior works (Jin et al., 2024; Chen et al., 2024a), we sequentially use each of ETTh1, ETTh2, ETTm1, and ETTm2 as the source dataset, treating the remaining datasets as targets.

The results in Table 5, measured on target datasets with a forecasting length of 192, highlight the consistent advantages of PS loss. It outperforms MSE loss in 34 out of 36 scenarios, demonstrating substantial improvements in generalization across diverse datasets and granularities. These gains arise from PS loss's ability to analyze time series at the patch level, incorporating local statistical properties to achieve better structural alignment and enabling models to adapt more effectively to unseen data distributions.

### 4.7. Forecasting Visualization

We visualize samples from the Weather dataset using iTransformer as the backbone model. As shown in Figure 3, PS loss significantly improves alignment with the ground truth compared to MSE loss: (a) its predictions closely follow the trajectory of the ground truth, preserving overall trends and patterns. (b) it effectively captures amplitude fluctuations, reflecting the localized variability. (c) it reduces biases at the overall level, ensuring better alignment with the central tendency of the ground truth. These results highlight the strengths of PS loss in capturing localized structural patterns. More visualizations are provided in Appendix H.

### 4.8. Hyperparameter Sensitivity

Our method introduces two key hyperparameters: PS loss weight $\lambda$ and patch length threshold $\delta$. We evaluate their impact using iTransformer model.

*Table 5.* Zero-shot forecasting results on ETT datasets. The table reports MSE and MAE values for backbone models trained with and without PS for zero-shot forecasting. The forecasting length is 192.

| Models | | iTransformer | | | | DLinear | | | | TimeMixer | | | |
|---|---|---|---|---|---|---|---|---|---|---|---|---|---|
| Loss Function | | MSE | | + PS | | MSE | | + PS | | MSE | | + PS | |
| Source | Target | MSE | MAE | MSE | MAE | MSE | MAE | MSE | MAE | MSE | MAE | MSE | MAE |
| ETTh1 | ETTh2 | **0.379** | 0.395 | 0.381 | **0.394** | 0.464 | 0.465 | **0.356** | **0.394** | 0.378 | 0.391 | **0.378** | **0.390** |
| | ETTm1 | 0.963 | 0.611 | **0.798** | **0.575** | 0.737 | 0.571 | **0.717** | **0.550** | 1.015 | 0.633 | **0.923** | **0.606** |
| | ETTm2 | 0.299 | 0.353 | **0.280** | **0.338** | 0.383 | 0.436 | **0.273** | **0.353** | 0.290 | 0.348 | **0.285** | **0.343** |
| ETTh2 | ETTh1 | 0.617 | 0.538 | **0.541** | **0.498** | 0.446 | 0.446 | **0.436** | **0.437** | 0.667 | 0.552 | **0.645** | **0.541** |
| | ETTm1 | 0.924 | 0.607 | **0.915** | **0.599** | **0.713** | **0.551** | 0.732 | 0.559 | 1.081 | 0.635 | **0.991** | **0.619** |
| | ETTm2 | 0.292 | 0.350 | **0.291** | **0.349** | 0.307 | 0.388 | **0.268** | **0.346** | 0.296 | 0.353 | **0.296** | **0.352** |
| ETTm1 | ETTh1 | 0.718 | 0.564 | **0.601** | **0.516** | 0.479 | 0.459 | **0.472** | **0.453** | 0.774 | 0.576 | **0.628** | **0.524** |
| | ETTh2 | 0.443 | 0.436 | **0.435** | **0.427** | 0.375 | 0.411 | **0.371** | **0.407** | 0.492 | 0.459 | **0.466** | **0.443** |
| | ETTm2 | 0.260 | 0.314 | **0.260** | **0.310** | 0.244 | 0.322 | **0.239** | **0.315** | 0.262 | 0.315 | **0.257** | **0.307** |
| ETTm2 | ETTh1 | 0.914 | 0.637 | **0.613** | **0.523** | 0.542 | 0.498 | **0.498** | **0.470** | 0.622 | 0.530 | **0.590** | **0.514** |
| | ETTh2 | 0.447 | 0.445 | **0.426** | **0.421** | 0.385 | 0.410 | **0.366** | **0.397** | 0.415 | 0.424 | **0.412** | **0.417** |
| | ETTm1 | 0.674 | 0.525 | **0.562** | **0.476** | 0.417 | 0.420 | **0.375** | **0.394** | 0.659 | 0.514 | **0.523** | **0.460** |

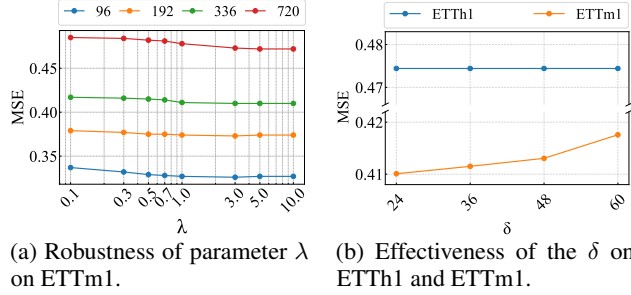

| (a) Better Correlation | (b) Better Variance Match | (c) Better Mean Match |
|---|---|---|

*Figure 3.* Forecasting visualization comparing PS loss and MSE loss as objective functions. Predictions with PS loss align more closely with the ground truth's statistical properties, yielding results with higher structural similarity.

(a) Robustness of parameter $\lambda$ on ETTm1.  (b) Effectiveness of the $\delta$ on ETTh1 and ETTm1.

*Figure 4.* The impact of the hyperparameter on ETTh1 and ETTm1 based iTransformer. More results on parameter $\lambda$ are shown in Appendix F.

**PS loss weight $\lambda$.** The weight $\lambda$ determines the relative importance of PS loss in combination with the MSE. We evaluate a range of $\lambda \in \{0.1, 0.3, 0.5, 0.7, 1.0, 3.0, 5.0, 10.0\}$, as depicted in Figure 4a. The results demonstrate that performance improves as $\lambda$ increases, peaking at $\lambda = 3.0$. Notably, the performance remains stable across a broad range of $\lambda$, indicating that PS loss is robust and can be easily applied to various datasets and models.

**Patch length threshold $\delta$.** The patch length threshold $\delta$ limits the maximum patch length to maintain the granularity of comparison. We evaluate $\delta \in \{24, 36, 48, 60\}$ and present the results in Figure 4b. For ETTh1, where the Fourier-based patch length is smaller than $\delta$, increasing $\delta$ does not affect performance. However, for ETTm1, where the Fourier-based patch length exceeds $\delta$, large $\delta$ results in diminished performance, indicating that overly large patches blur fine-grained structural patterns, and proper patch length is crucial for structural alignment.

## 5. Conclusion

In this study, we addressed the limitations of traditional point-wise loss functions by introducing the Patch-wise Structural (PS) loss, a novel method that leverages local statistical properties for more precise structural alignment. By integrating patch-level analysis and dynamic weighting strategies, PS loss improves forecasting accuracy and strengthens generalization across multivariate time series models. This innovation establishes a new benchmark for accurately modeling complex time series data. In future work, we will focus on developing a new loss function and model that accounts for both local and global structural dynamics within the prediction.

## Acknowledgments

This work was supported by the National Natural Science Foundation of China under Grants 62376194, 61925602, U23B2049 and China Scholarship Council Grant 202406250137.

## Impact Statement

This paper presents work whose goal is to advance the field of Machine Learning. There are many potential societal consequences of our work, none which we feel must be specifically highlighted here.

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

# A. Datasets

Our study leverages a diverse set of forecasting datasets to evaluate the effectiveness of our loss function across various domains.

- **ETT** (Electricity Transformer Temperature): The ETT (Zhou et al., 2021) dataset contains seven features of electricity transformer data collected from two separate counties between July 2016 and July 2018. These datasets vary in granularity, with "h" indicating hourly data and "m" indicating 15-minute intervals. The suffixes "1" and "2" refer to two different regions from which the datasets originated.

- **Weather**: The Weather (Zhou et al., 2021) dataset includes 21 meteorological factors recorded every 10 minutes in 2020 at the Max Planck Biogeochemistry Institute's Weather Station.

- **ECL** (Electricity Consuming Load): The ECL (Li et al., 2019) dataset contains hourly electricity consumption data of 321 clients from 2012 to 2014.

- **Exchange**: The Exchange (Lai et al., 2018) dataset consists of daily exchange rates from 1990 to 2016 for eight countries: Australia, the United Kingdom, Canada, Switzerland, China, Japan, New Zealand, and Singapore.

We follow the same data processing and train-validation-test split protocol as previous works (Zhou et al., 2021; Wu et al., 2023a; Liu et al., 2024a). The ETT dataset is divided into training, validation, and testing sets in a 12/4/4-month ratio, while other datasets are split in a 7:1:2 ratio. The details of the datasets are provided in Table 6.

Table 6. Descriptions of the forecasting datasets. "Channel" shows the variate number of each dataset. "Dataset Size" indicates the size of the (Train, Validation, and Test) split respectively. "Frequency" denotes the sampling interval of time points.

| Dataset | Channel | Length | Datset Size | Frequency | Information |
|---------|---------|--------|-------------|-----------|-------------|
| ETTh1 | 7 | 17420 | (8545, 2881, 2881) | 1 hour | Electricity |
| ETTh2 | 7 | 17420 | (8545, 2881, 2881) | 1 hour | Electricity |
| ETTm1 | 7 | 69680 | (34465, 11521, 11521) | 15 min | Electricity |
| ETTm2 | 7 | 69680 | (34465, 11521, 11521) | 15 min | Electricity |
| Weather | 21 | 52696 | (36792, 5271, 10540) | 10 min | Weather |
| ECL | 321 | 26304 | (18317, 2633, 5261) | 1 hour | Electricity |
| Exchange | 8 | 7588 | (5120, 665, 1422) | 1 day | Exchange Rates |

# B. Backbone Models

We select seven state-of-the-art (SOTA) models from different architectures as backbones. These include Transformer-based models: iTransformer (Liu et al., 2024a) and PatchTST (Nie et al., 2023); MLP-based models: DLinear (Zeng et al., 2023) and TimeMixer (Wang et al., 2024); CNN-based model: TimesNet (Wu et al., 2023a); LLM-based models: OFA (Zhou et al., 2023) and AutoTimes (Liu et al., 2024b). We use the official implementations of each backbone from their respective repositories and follow their optimal hyperparameter settings for our experiments.

- iTransformer[1]: It embeds the entire series as a single token and applies a Transformer along the channel dimension to explicitly capture channel dependencies.

- PatchTST[2]: It segments time series into patches and applies a Transformer along the time dimension to capture long-term temporal dependencies.

- DLinear[3]: It simplifies forecasting by decomposing time series into trend and seasonal components, modeling each separately with linear layers.

---

[1] https://github.com/thuml/iTransformer
[2] https://github.com/yuqinie98/PatchTST
[3] https://github.com/cure-lab/LTSF-Linear

- TimeMixer[4]: It captures multi-scale information by decomposing time series into different scales and blending them through linear layers.

- TimesNet[5]: It transforms 1D time series into 2D tensors to account for multi-periodicity and employs CNNs to capture both inter- and intra-period dependencies.

- OFA[6]: It embeds time series patches into tokens and leverages pre-trained language models for general time series analysis tasks.

- AutoTimes[7]: It projects time series into the embedding space of language tokens and autoregressively generates future predictions with arbitrary lengths.

## C. Related Work

**Patching Mechanism in Time Series Forecasting.** In recent years, patch-based approaches have gained significant traction in time series forecasting, offering substantial improvements in prediction performance (Zhang & Yan, 2023; Chen et al., 2024b). By segmenting the input time series into patches, these methods enhance locality and capture comprehensive semantic information that point-level techniques often overlook (Nie et al., 2023; Zhang & Yan, 2023).

For instance, PatchTST (Nie et al., 2023) employs series patching combined with a channel-independent (CI) modeling strategy to capture long-term temporal dependencies. Crossformer (Zhang & Yan, 2023) partitions time series into patches and utilizes self-attention mechanisms to model dependencies across both temporal and variable dimensions. TimesNet (Wu et al., 2023a) and PDF (Dai et al., 2024) apply Fast Fourier Transform (FFT) to determine the dominant frequency of the input series, segmenting the data accordingly to ensure patches capture clear periodic patterns. HDMixer (Huang et al., 2024b) adaptively expands patch lengths to enrich boundary information and mitigate semantic incoherence within the series.

While these methods employ patching to extract localized semantic information from input time series, we take a novel approach by introducing patching to the prediction series. This enables a localized comparison between prediction and ground truth, providing a more fine-grained assessment of structural similarity.

## D. Computational Cost

To evaluate the computational cost introduced by the proposed PS loss, we conducted a theoretical complexity analysis and measured the actual runtime overhead during training.

### D.1. Time Complexity Analysis

We analyze the complexity of our proposed PS loss with respect to the forecasting length $T$, channel number $C$, and the last hidden layer's dimension $d$. The overall time complexity comes from three main components:

- **Fourier-based Adaptive Patching (FAP).** The time complexity for this component is dominated by the Fast Fourier Transform (FFT) operation, which has a complexity of $O(T \log T)$ for a time series of length T. Since the FFT is applied independently to each of the C channels, the total time complexity is $O(C \cdot T \log T)$.

- **Patch-wise Structural Loss (PS).** The series is divided into $N = \lfloor \frac{(T-P)}{S} \rfloor + 1 \approx \frac{2T}{P}$ patches, where $P$ denotes the patch length and the stride $S$ is set to $\lfloor \frac{P}{2} \rfloor$. Computing the correlation, variance, and mean for each patch requires $P$ operations. With a total of $C \cdot N$ patches, the time complexity of this component is $O(C \cdot N \cdot P) = O(C \cdot T)$.

- **Gradient-based Dynamic Weighting (GDW).** The gradient computation for each loss term with respect to the output layer weights, which have shape $d \times T$, requires $O(d \cdot T)$ operations. Considering all three loss components, the total time complexity of GDW is $O(3 \cdot d \cdot T) = O(d \cdot T)$.

---

[4] https://github.com/kwuking/TimeMixer
[5] https://github.com/thuml/Time-Series-Library
[6] https://github.com/DAMO-DI-ML/NeurIPS2023-One-Fits-All
[7] https://github.com/thuml/AutoTimes

Therefore, the overall time complexity of the PS loss is $O(C \cdot T \log T + C \cdot T + d \cdot T)$.

### D.2. Actual Runtime Overhead

To assess the runtime overhead introduced by the PS loss, we measure the training time per epoch (seconds/epoch) for both the standard MSE loss and the proposed PS loss, using the iTransformer model as the backbone. We select three datasets of different scales for the experiment: ETTh1 (small), Weather (medium), and ECL (large). As shown in Table 7, while PS loss introduces additional computational cost, the increase in training time remains modest across datasets of different sizes. This overhead is justified by the consistent improvements in forecasting accuracy achieved with the PS loss.

*Table 7.* Runtime Overhead of PS loss. The table reports the running time per epoch (seconds/epoch) for iTransformer model using MSE loss and PS loss on the three datasets: ETTh1, Weather, and ECL. "Time Inc." represents the additional running time introduced by PS loss.

| Datasets | | ETTh1 | | | Weather | | | ECL | | |
|---|---|---|---|---|---|---|---|---|---|---|
| Train Time (s/epoch) | | MSE Time | PS Time | *Time Inc.* | MSE Time | PS Time | *Time Inc.* | MSE Time | PS Time | *Time Inc.* |
| iTransformer | 96 | 1.99 | 2.64 | 0.65 | 10.25 | 13.27 | 3.03 | 24.03 | 27.36 | 3.34 |
| | 192 | 1.98 | 2.73 | 0.74 | 10.62 | 14.17 | 3.55 | 24.35 | 28.23 | 3.89 |
| | 336 | 1.91 | 2.64 | 0.73 | 10.71 | 14.52 | 3.81 | 25.15 | 30.13 | 4.98 |
| | 720 | 1.95 | 2.65 | 0.70 | 10.96 | 14.19 | 3.23 | 26.55 | 35.07 | 8.52 |
| | *Avg.* | 1.96 | 2.66 | 0.71 | 10.63 | 14.04 | 3.40 | 25.02 | 30.20 | 5.18 |

## E. Additional Evaluation Metrics for PS Loss Performance

To provide a more comprehensive evaluation of the PS loss, we incorporate additional metrics, including Dynamic Time Warping (DTW) (Müller, 2007), Time Distortion Index (TDI) (Frías-Paredes et al., 2016), and Pearson Correlation Coefficient (PCC) (Cohen et al., 2009), to assess the quality of the forecasting results.

### E.1. Metric Definitions

**Dynamic Time Warping.** DTW computes the distance between two time series by identifying an optimal warping path, allowing non-linear alignments along the time axis to emphasize shape similarity (Müller, 2007). Given two time series $Y = \{y_0, y_1, \ldots, y_{T-1}\} \in \mathbb{R}^T$ and $\hat{Y} = \{\hat{y}_0, \hat{y}_1, \ldots, \hat{y}_{T-1}\} \in \mathbb{R}^T$, the DTW distance is defined as:

$$\mathrm{DTW}(Y, \hat{Y}) = \min_{\mathbf{A} \in \mathcal{A}(Y, \hat{Y})} \sum_{(i,j) \in \mathbf{A}} d(y_i, \hat{y}_j) = \sum_{(i,j) \in \mathbf{A}^*} d(y_i, \hat{y}_j),$$

where $d(\cdot, \cdot)$ denotes a distance measure, typically the squared Euclidean distance. $\mathbf{A}$ is a warping path consisting of $K$ index pairs $\{(i_0, j_0), (i_1, j_1), \ldots, (i_{K-1}, j_{K-1})\}$, where $0 \leq i_k, j_k \leq T - 1$. $\mathcal{A}(Y, \hat{Y})$ denotes the set of all admissible warping paths, and $\mathbf{A}^* \in \mathcal{A}(Y, \hat{Y})$ is the optimal path that yields the minimum total distance between aligned time steps. A warping path $\mathbf{A}$ is considered admissible if it satisfies the following conditions:

- Boundary condition: $(i_0, j_0) = (0, 0)$ and $(i_{K-1}, j_{K-1}) = (T - 1, T - 1)$
- Monotonicity condition: $i_{k+1} \geq i_k$ and $j_{k+1} \geq j_k$, for all $k \in [0, K - 2]$
- Step size condition: $(i_{k+1} - i_k, j_{k+1} - j_k) \in \{(1, 0), (0, 1), (1, 1)\}$, for all $k \in [0, K - 2]$

**Time Distortion Index.** TDI quantifies the degree of temporal distortion between two sequences (Frías-Paredes et al., 2016). It was originally defined as the area between the optimal DTW warping path $\mathbf{A}^*$ and the identity path $\mathcal{I} = \{(t, t)\}_{t=0}^{T-1}$. In our work, we adopt a simplified version of TDI proposed in (Le Guen & Thome, 2019). Given two time series $Y \in \mathbb{R}^T$ and $\hat{Y} \in \mathbb{R}^T$ and their optimal warping path $\mathbf{A}^*$, the TDI is calculated as:

$$\mathrm{TDI}(Y, \hat{Y}) = \sum_{(i,j) \in \mathbf{A}^*} \frac{(i - j)^2}{T^2}.$$

**Pearson Correlation Coefficient.** PCC measures the degree of linear correlation between two time series. Given $Y = \{y_0, y_1, \ldots, y_{T-1}\} \in \mathbb{R}^T$ and $\hat{Y} = \{\hat{y}_0, \hat{y}_1, \ldots, \hat{y}_{T-1}\} \in \mathbb{R}^T$, the PCC is calculated as:

$$\text{PCC}(Y, \hat{Y}) = \frac{\sum_{t=0}^{T-1}(y_t - \bar{y})(\hat{y}_t - \bar{\hat{y}})}{\sqrt{\sum_{t=0}^{T-1}(y_t - \bar{y})^2} \cdot \sqrt{\sum_{t=0}^{T-1}(\hat{y}_t - \bar{\hat{y}})^2}},$$

where $\bar{y}$ and $\bar{\hat{y}}$ denote the mean values of $Y$ and $\hat{Y}$, respectively.

### E.2. Evaluation Results

We evaluated the forecasting performance of the iTransformer model trained with MSE and PS loss using DTW, TDI, and PCC. The results are presented in Table 8. PS loss consistently improves all three shape-aware metrics, indicating better structural alignment. On the ETTh2 dataset, the iTransformer trained with MSE achieves lower DTW scores, reflecting smaller numerical differences after optimal alignment. However, the higher TDI in this case suggests that the alignment involves more extensive temporal warping, indicating greater structural distortion compared to the forecasts generated using PS loss.

*Table 8.* Additional evaluation metrics for assessing the performance of PS loss. This table reports the performance of the iTransformer model trained with MSE and PS loss, evaluated using three shape-aware metrics: DTW, TDI, and PCC.

| Model | | iTransformer | | | | | | | |
|---|---|---|---|---|---|---|---|---|---|
| Metrics | | MSE | | DTW | | TDI | | PCC | |
| Loss Functions | | MSE | + PS | MSE | + PS | MSE | + PS | MSE | + PS |
| ETTh1 | 96 | 0.387 | **0.379** | 3.743 | **3.725** | 2.475 | **2.278** | 0.562 | **0.572** |
| | 192 | 0.441 | **0.428** | 5.728 | **5.685** | 4.549 | **4.479** | 0.530 | **0.539** |
| | 336 | 0.491 | **0.474** | 8.055 | **8.042** | 9.356 | **8.393** | 0.496 | **0.519** |
| | 720 | 0.509 | **0.480** | 11.896 | **11.844** | 15.171 | **12.686** | 0.469 | **0.488** |
| | Avg | 0.457 | **0.440** | 7.355 | **7.324** | 7.888 | **6.959** | 0.514 | **0.530** |
| ETTh2 | 96 | 0.301 | **0.289** | 3.256 | **3.213** | 5.739 | **5.207** | 0.355 | **0.408** |
| | 192 | 0.380 | **0.373** | **5.227** | 5.321 | 14.124 | **13.937** | 0.311 | **0.353** |
| | 336 | 0.424 | **0.416** | **7.494** | 7.644 | 27.298 | **24.994** | 0.286 | **0.325** |
| | 720 | 0.430 | **0.421** | **11.586** | 11.886 | 51.733 | **46.683** | 0.244 | **0.283** |
| | Avg | 0.384 | **0.375** | **6.891** | 7.016 | 24.723 | **22.705** | 0.299 | **0.342** |
| ETTm1 | 96 | 0.342 | **0.326** | 3.275 | **3.167** | 6.199 | **5.569** | 0.575 | **0.604** |
| | 192 | 0.383 | **0.374** | 4.887 | **4.796** | 8.221 | **7.561** | 0.556 | **0.573** |
| | 336 | 0.418 | **0.410** | 6.917 | **6.815** | 11.863 | **10.831** | 0.530 | **0.545** |
| | 720 | 0.487 | **0.472** | 11.195 | **10.960** | 23.522 | **21.562** | 0.490 | **0.505** |
| | Avg | 0.408 | **0.396** | 6.568 | **6.435** | 12.451 | **11.381** | 0.538 | **0.557** |
| ETTm2 | 96 | 0.186 | **0.175** | 2.575 | **2.358** | 6.338 | **5.544** | 0.364 | **0.425** |
| | 192 | 0.254 | **0.242** | 4.141 | **3.906** | 11.684 | **10.748** | 0.337 | **0.403** |
| | 336 | 0.316 | **0.304** | 6.167 | **5.878** | 25.611 | **21.580** | 0.318 | **0.377** |
| | 720 | 0.414 | **0.401** | 10.770 | **10.300** | 64.242 | **52.106** | 0.281 | **0.341** |
| | Avg | 0.292 | **0.281** | 5.913 | **5.611** | 26.969 | **22.495** | 0.325 | **0.387** |
| Weather | 96 | 0.176 | **0.167** | 2.174 | **2.107** | 11.990 | **11.166** | 0.344 | **0.381** |
| | 192 | 0.227 | **0.219** | 3.725 | **3.703** | 22.741 | **21.400** | 0.339 | **0.371** |
| | 336 | 0.282 | **0.274** | 5.768 | 5.772 | 42.628 | **41.029** | 0.319 | **0.344** |
| | 720 | 0.357 | **0.353** | **9.975** | 10.053 | 88.401 | **87.777** | 0.296 | **0.313** |
| | Avg | 0.261 | **0.253** | 5.410 | **5.409** | 41.440 | **40.343** | 0.324 | **0.352** |

## F. More Results on Hyperparameter Sensitivity

We investigate the impact of hyperparameter PS loss weight $\lambda$ on ETTm2 and Weather datasets using iTransformer and DLinear as the backbone models. We set $\lambda \in \{0.1, 0.3, 0.5, 0.7, 1.0, 3.0, 5.0, 10.0\}$ for the experiment.

The results show that as $\lambda$ increases, the performance of the model improves, with $\lambda = 5.0$ achieving the relatively best results. Moreover, the performance remains robust across a wide range of $\lambda$ values tested, which simplifies PS loss application to different datasets and models.

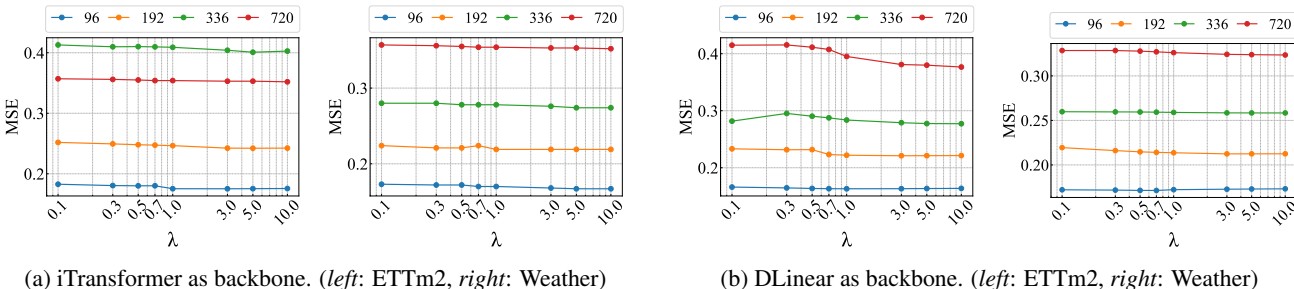

(a) iTransformer as backbone. (*left*: ETTm2, *right*: Weather)  (b) DLinear as backbone. (*left*: ETTm2, *right*: Weather)

*Figure 5.* Influence of PS loss weight $\lambda$ on iTransformer and DLinear backbone. We plot the change in MSE with respect to $\lambda$ on the ETTm2 and Weather datasets, using iTransformer and DLinear as the backbone model. The performance is robust across a wide range of $\lambda$ values.

## G. Impact of PS Loss on Generalization

To examine how PS loss affects training dynamics and generalization, we visualize the MSE on both the training and test datasets across all epochs, using MSE loss and PS loss as objective functions, as shown in Figure 6. The experiments were conducted on the ETTh1, ETTh2, ETTm1, and ETTm2 datasets, with iTransformer as the backbone model.

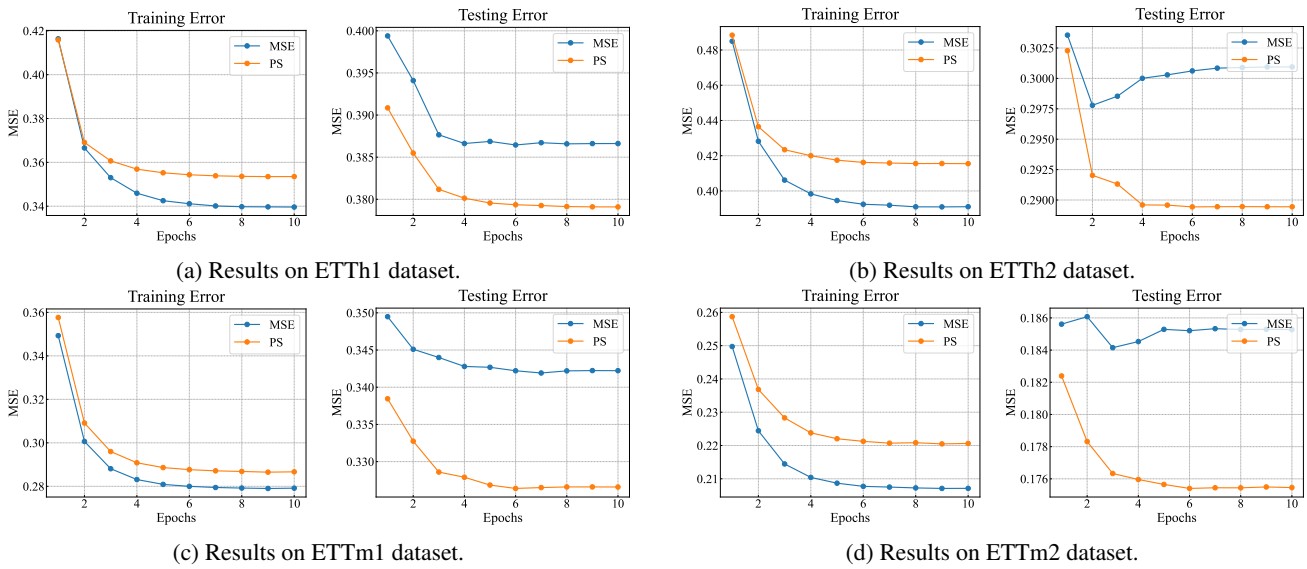

(a) Results on ETTh1 dataset.          (b) Results on ETTh2 dataset.

(c) Results on ETTm1 dataset.          (d) Results on ETTm2 dataset.

*Figure 6.* Training and testing MSE loss curves across all training epochs for the iTransformer model trained with MSE loss and PS loss on the ETT datasets (*left*: Training errors, *right*: Testing errors). The model trained with PS loss shows higher training errors but achieves lower testing errors, highlighting its effectiveness in enhancing generalization and mitigating overfitting.

From Figure 6, we observe a consistent trend across both datasets. During training, models optimized solely with MSE loss achieve lower training error per epoch compared to those optimized with PS loss. However, on testing data, models trained with MSE loss exhibit higher testing error per epoch compared to those trained with PS loss. These observations suggest that models trained with MSE loss focus solely on the point-wise error, resulting in smaller training losses, but fail to generalize effectively to the testing data. In contrast, PS loss mitigates overfitting by encouraging the model to focus on local structural similarity in addition to point-wise accuracy, highlighting its effectiveness in improving generalization and forecasting accuracy.

# H. Forecasting Visualization

To evaluate the quality of predictions using MSE loss and our proposed PS loss, we visualize samples from the ETTm1, ETTm2, and Weather datasets with iTransformer as the backbone model. Predictions generated with PS loss exhibit enhanced alignment with the ground truth, accurately capturing directionality, fluctuation dynamics, and average levels while preserving localized patterns and structural details.

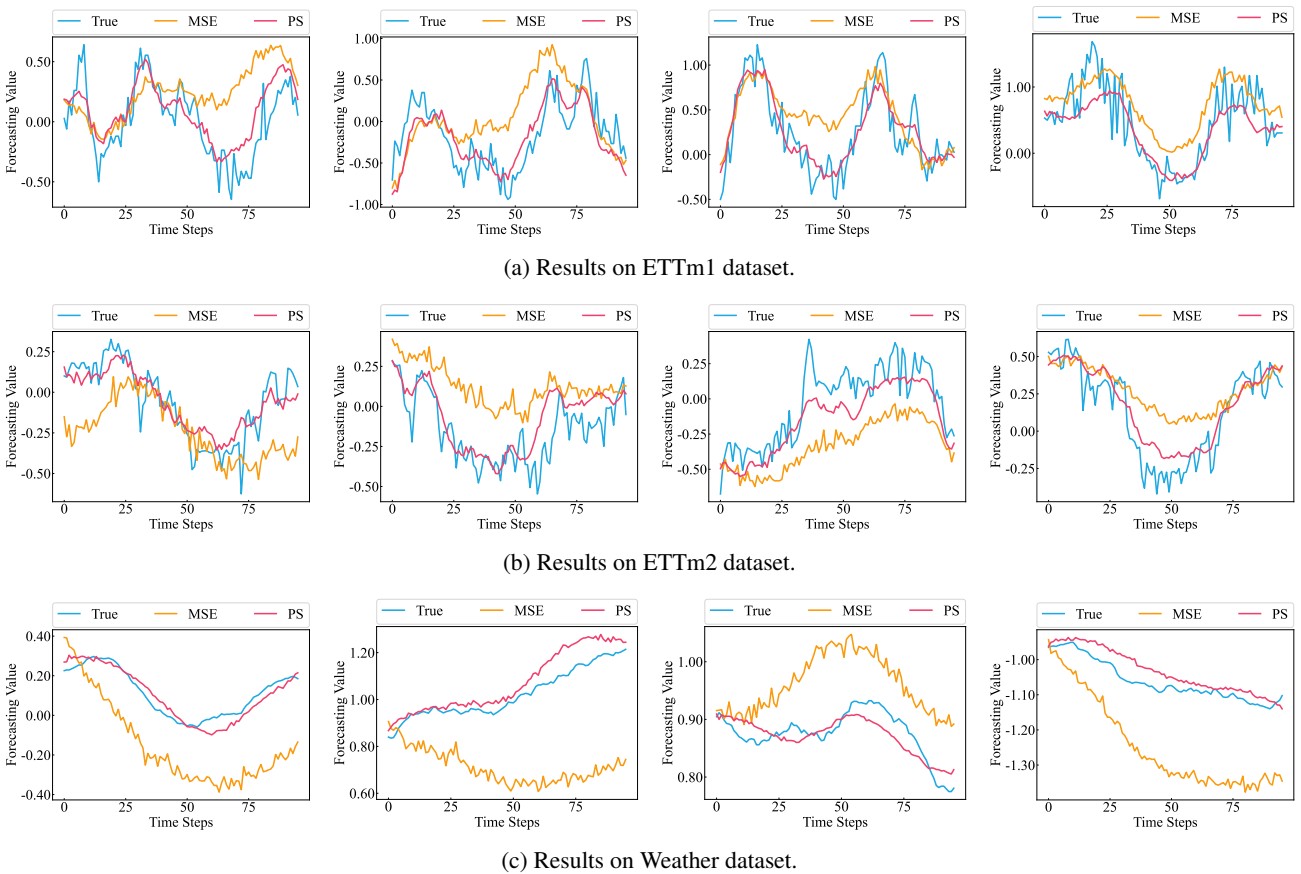

(a) Results on ETTm1 dataset.

(b) Results on ETTm2 dataset.

(c) Results on Weather dataset.

*Figure 7.* Forecasting visualization comparing PS loss and MSE loss as objective functions.

# I. More Results on Ablation Study

We conduct additional ablation studies on the DLinear and iTransformer backbones using the ETTh1, ETTh2, ETTm1, and ETTm2 datasets to further demonstrate the necessity of each component, as shown in Table 9 and Table 10.

- **Loss components.** Omitting any one of the three loss components results in a decline in performance, underscoring the unique and complementary roles each loss plays in capturing localized structural similarities.

- **Adaptive Patching.** Comparing time series globally leads to performance degradation, especially for longer forecasting horizons. This highlights the importance of point-wise comparison in capturing nuanced local variations in statistical properties.

- **Dynamic weighting.** Replacing the GDW strategy with fixed weighting leads to performance degradation, highlighting the importance of GDW in ensuring the balanced utilization of each loss term for effective model training.

*Table 9.* Ablation study of the components of PS loss on the ETT datasets using DLinear as a backbone.

| Method | | Dlinear + PS | | w/o $\mathcal{L}_{Corr}$ | | w/o $\mathcal{L}_{Var}$ | | w/o $\mathcal{L}_{Mean}$ | | w/o Patching | | w/o Weighting | |
|---|---|---|---|---|---|---|---|---|---|---|---|---|---|
| Metric | | MSE | MAE | MSE | MAE | MSE | MAE | MSE | MAE | MSE | MAE | MSE | MAE |
| ETTh1 | 96 | **0.367** | **0.389** | 0.381 | 0.402 | 0.380 | 0.399 | 0.372 | 0.395 | 0.380 | 0.401 | 0.384 | 0.404 |
| | 192 | **0.402** | **0.411** | 0.439 | 0.446 | 0.440 | 0.444 | 0.404 | 0.414 | 0.436 | 0.441 | 0.403 | 0.411 |
| | 336 | **0.435** | **0.435** | 0.443 | 0.445 | 0.439 | 0.486 | 0.440 | 0.443 | 0.442 | 0.442 | 0.439 | 0.440 |
| | 720 | **0.463** | **0.484** | 0.510 | 0.519 | 0.509 | 0.517 | 0.474 | 0.493 | 0.522 | 0.528 | 0.528 | 0.532 |
| | Avg | **0.417** | **0.430** | 0.443 | 0.453 | 0.442 | 0.462 | 0.423 | 0.436 | 0.445 | 0.453 | 0.439 | 0.447 |
| ETTh2 | 96 | **0.280** | **0.341** | 0.289 | 0.351 | 0.281 | 0.342 | 0.287 | 0.351 | 0.280 | 0.341 | 0.283 | 0.345 |
| | 192 | **0.358** | **0.396** | 0.381 | 0.415 | 0.359 | 0.396 | 0.397 | 0.428 | 0.386 | 0.399 | 0.371 | 0.404 |
| | 336 | **0.435** | **0.450** | 0.455 | 0.466 | 0.457 | 0.462 | 0.477 | 0.477 | 0.438 | 0.450 | 0.439 | 0.452 |
| | 720 | **0.597** | **0.540** | 0.720 | 0.599 | 0.673 | 0.574 | 0.747 | 0.612 | 0.663 | 0.583 | 0.679 | 0.580 |
| | Avg | **0.417** | **0.432** | 0.461 | 0.458 | 0.443 | 0.443 | 0.477 | 0.467 | 0.442 | 0.443 | 0.443 | 0.445 |
| ETTm1 | 96 | **0.296** | 0.339 | 0.300 | 0.343 | **0.296** | **0.338** | 0.301 | 0.345 | 0.304 | 0.347 | 0.297 | 0.340 |
| | 192 | 0.333 | 0.362 | 0.334 | 0.364 | **0.332** | **0.359** | 0.335 | 0.366 | 0.341 | 0.373 | 0.332 | 0.361 |
| | 336 | **0.365** | **0.380** | 0.370 | 0.387 | 0.368 | 0.381 | 0.369 | 0.386 | 0.371 | 0.387 | 0.367 | 0.381 |
| | 720 | **0.419** | **0.413** | 0.424 | 0.420 | 0.422 | 0.415 | 0.423 | 0.420 | 0.423 | 0.419 | 0.422 | 0.415 |
| | Avg | **0.353** | **0.374** | 0.357 | 0.378 | 0.355 | 0.373 | 0.357 | 0.379 | 0.360 | 0.381 | 0.355 | 0.374 |
| ETTm2 | 96 | **0.163** | **0.251** | 0.170 | 0.265 | 0.163 | 0.252 | 0.171 | 0.268 | 0.164 | 0.254 | 0.164 | 0.255 |
| | 192 | **0.222** | **0.296** | 0.234 | 0.308 | 0.222 | 0.296 | 0.238 | 0.320 | 0.223 | 0.299 | 0.225 | 0.301 |
| | 336 | **0.277** | **0.332** | 0.282 | 0.342 | 0.283 | 0.342 | 0.295 | 0.353 | 0.286 | 0.347 | 0.289 | 0.350 |
| | 720 | **0.377** | **0.397** | 0.414 | 0.432 | 0.398 | 0.418 | 0.451 | 0.453 | 0.393 | 0.415 | 0.410 | 0.426 |
| | Avg | **0.260** | **0.319** | 0.275 | 0.337 | 0.267 | 0.327 | 0.289 | 0.348 | 0.266 | 0.329 | 0.272 | 0.333 |

*Table 10.* Ablation study of the components of PS loss on the ETT datasets using iTransformer as a backbone. "iTrans∗" refers to the iTransformer model.

| Method | | iTrans∗ + PS | | w/o $\mathcal{L}_{Corr}$ | | w/o $\mathcal{L}_{Var}$ | | w/o $\mathcal{L}_{Mean}$ | | w/o Patching | | w/o Weighting | |
|---|---|---|---|---|---|---|---|---|---|---|---|---|---|
| Metric | | MSE | MAE | MSE | MAE | MSE | MAE | MSE | MAE | MSE | MAE | MSE | MAE |
| ETTh1 | 96 | **0.379** | **0.396** | 0.381 | 0.399 | 0.385 | 0.400 | 0.382 | 0.397 | 0.383 | 0.400 | 0.380 | 0.396 |
| | 192 | **0.428** | **0.424** | 0.441 | 0.433 | 0.436 | 0.429 | 0.431 | 0.426 | 0.435 | 0.429 | 0.430 | 0.425 |
| | 336 | 0.474 | 0.453 | 0.486 | 0.458 | 0.499 | 0.466 | **0.471** | **0.446** | 0.483 | 0.455 | 0.485 | 0.458 |
| | 720 | 0.480 | 0.478 | 0.515 | 0.496 | 0.521 | 0.498 | **0.477** | **0.471** | 0.511 | 0.492 | 0.532 | 0.505 |
| | Avg | **0.440** | 0.438 | 0.456 | 0.447 | 0.460 | 0.448 | **0.440** | **0.435** | 0.453 | 0.444 | 0.457 | 0.446 |
| ETTh2 | 96 | **0.289** | **0.340** | 0.294 | 0.344 | 0.296 | 0.343 | 0.292 | 0.343 | 0.290 | 0.340 | 0.292 | 0.341 |
| | 192 | **0.373** | **0.391** | 0.374 | 0.393 | 0.375 | 0.392 | 0.379 | 0.395 | 0.376 | 0.393 | 0.374 | 0.392 |
| | 336 | **0.416** | **0.428** | 0.423 | 0.431 | 0.421 | 0.429 | 0.424 | 0.431 | 0.419 | 0.428 | 0.419 | 0.428 |
| | 720 | **0.421** | **0.441** | 0.439 | 0.453 | 0.425 | 0.442 | 0.426 | 0.443 | 0.423 | 0.441 | 0.423 | 0.441 |
| | Avg | **0.375** | **0.400** | 0.383 | 0.405 | 0.379 | 0.402 | 0.380 | 0.403 | 0.377 | 0.401 | 0.377 | 0.401 |
| ETTm1 | 96 | **0.326** | **0.360** | 0.334 | 0.365 | 0.328 | 0.360 | 0.350 | 0.375 | 0.343 | 0.374 | 0.328 | 0.361 |
| | 192 | **0.374** | **0.384** | 0.382 | 0.387 | 0.374 | 0.381 | 0.384 | 0.390 | 0.385 | 0.394 | 0.375 | 0.383 |
| | 336 | **0.410** | **0.406** | 0.419 | 0.412 | 0.412 | 0.406 | 0.417 | 0.412 | 0.420 | 0.416 | 0.411 | 0.407 |
| | 720 | **0.472** | **0.440** | 0.482 | 0.446 | 0.479 | 0.443 | 0.483 | 0.449 | 0.488 | 0.455 | 0.477 | 0.444 |
| | Avg | **0.396** | **0.397** | 0.404 | 0.402 | 0.398 | 0.397 | 0.408 | 0.407 | 0.409 | 0.410 | 0.398 | 0.399 |
| ETTm2 | 96 | **0.175** | **0.254** | 0.179 | 0.259 | 0.177 | 0.256 | 0.183 | 0.266 | 0.182 | 0.265 | 0.177 | 0.258 |
| | 192 | **0.242** | **0.299** | 0.248 | 0.306 | 0.244 | 0.300 | 0.247 | 0.306 | 0.248 | 0.306 | 0.245 | 0.302 |
| | 336 | **0.304** | **0.338** | 0.309 | 0.343 | 0.308 | 0.340 | 0.308 | 0.344 | 0.310 | 0.344 | 0.309 | 0.343 |
| | 720 | **0.401** | **0.394** | 0.409 | 0.400 | 0.405 | 0.397 | 0.410 | 0.401 | 0.410 | 0.400 | 0.410 | 0.399 |
| | Avg | **0.281** | **0.321** | 0.286 | 0.327 | 0.283 | 0.323 | 0.287 | 0.329 | 0.287 | 0.329 | 0.285 | 0.325 |

## J. More Experiments on Zero-shot Forecasting

**Different Forecasting Lengths.** We conducted zero-shot forecasting experiments on the iTransformer model with forecasting lengths of 96, 336, and 720. As results shown in Table 11, PS loss consistently enhances forecasting accuracy across all horizons, yielding improvements in 33 out of 36 settings. These findings underscore the effectiveness of PS loss in improving zero-shot forecasting performance under varying prediction lengths.

*Table 11.* PS loss zero-shot performance on different forecasting lengths using iTransformer as the backbone model. The forecasting lengths are {96, 336, 720}.

| Forecasting Length | | 96 | | | | 336 | | | | 720 | | | |
| --- | --- | --- | --- | --- | --- | --- | --- | --- | --- | --- | --- | --- | --- |
| Loss Function | | MSE | | PS | | MSE | | PS | | MSE | | PS | |
| Source | Target | MSE | MAE | MSE | MAE | MSE | MAE | MSE | MAE | MSE | MAE | MSE | MAE |
| ETTh1 | ETTh2 | 0.296 | 0.344 | **0.296** | **0.342** | 0.424 | 0.431 | **0.422** | **0.425** | **0.429** | **0.444** | 0.432 | 0.445 |
| | ETTm1 | 0.963 | 0.611 | **0.821** | **0.570** | **0.845** | **0.597** | 0.949 | 0.616 | 0.866 | 0.610 | **0.828** | **0.601** |
| | ETTm2 | 0.238 | 0.321 | **0.222** | **0.307** | 0.342 | 0.375 | 0.353 | 0.382 | 0.442 | 0.428 | **0.436** | **0.424** |
| ETTh2 | ETTh1 | 0.575 | 0.515 | **0.465** | **0.455** | 0.656 | 0.559 | **0.607** | **0.534** | 0.708 | 0.603 | **0.626** | **0.563** |
| | ETTm1 | 1.066 | 0.641 | **0.927** | **0.597** | 0.918 | 0.615 | **0.887** | **0.603** | 0.906 | 0.629 | **0.869** | **0.614** |
| | ETTm2 | 0.237 | 0.321 | **0.228** | **0.313** | 0.349 | 0.381 | **0.347** | **0.380** | 0.445 | 0.430 | **0.441** | **0.428** |
| ETTm1 | ETTh1 | 0.708 | 0.556 | **0.617** | **0.521** | 0.731 | 0.576 | **0.626** | **0.532** | 0.743 | 0.599 | **0.633** | **0.559** |
| | ETTh2 | 0.351 | 0.390 | **0.342** | **0.380** | 0.490 | 0.469 | **0.473** | **0.457** | 0.486 | 0.476 | **0.467** | **0.463** |
| | ETTm2 | 0.202 | 0.279 | **0.198** | **0.272** | 0.322 | 0.352 | **0.319** | **0.347** | 0.422 | 0.407 | **0.419** | **0.403** |
| ETTm2 | ETTh1 | 0.833 | 0.603 | **0.568** | **0.497** | 1.119 | 0.716 | **0.642** | **0.539** | 1.202 | 0.756 | **0.623** | **0.544** |
| | ETTh2 | 0.353 | 0.395 | **0.341** | **0.373** | 0.517 | 0.491 | **0.461** | **0.451** | 0.528 | 0.505 | **0.456** | **0.460** |
| | ETTm1 | 0.679 | 0.520 | **0.465** | **0.425** | 0.740 | 0.557 | **0.520** | **0.464** | 0.880 | 0.615 | **0.584** | **0.500** |

**LLM-based Models.** To further assess the generalization ability of the proposed PS loss, we extend our zero-shot forecasting experiments to LLM-based models, including OFA (Zhou et al., 2023), AutoTimes (Liu et al., 2024b), and Time-LLM (Jin et al., 2024), with the forecasting length set to 96. As shown in Table 12, incorporating PS loss consistently improves forecasting performance in most settings. These results demonstrate the compatibility of PS loss with LLM-based models and highlight its ability to enhance generalization in zero-shot forecasting.

*Table 12.* PS loss performance on LLM-based models for the zero-shot forecasting task. The backbones are OFA and Autotimes, with a forecasting length of 96.

| Models | | OFA | | | | AutoTimes | | | | Time-LLM | | | |
| --- | --- | --- | --- | --- | --- | --- | --- | --- | --- | --- | --- | --- | --- |
| Loss Function | | MSE loss | | PS loss | | MSE loss | | PS loss | | MSE loss | | PS loss | |
| Source | Target | MSE | MAE | MSE | MAE | MSE | MAE | MSE | MAE | MSE | MAE | MSE | MAE |
| ETTh1 | ETTh2 | **0.289** | 0.347 | 0.292 | **0.343** | **0.303** | **0.362** | 0.305 | 0.363 | 0.287 | 0.349 | **0.286** | **0.346** |
| | ETTm1 | **0.723** | **0.532** | 0.742 | 0.534 | 0.728 | 0.552 | **0.724** | **0.549** | 0.747 | 0.566 | **0.706** | **0.538** |
| | ETTm2 | 0.219 | 0.311 | **0.216** | **0.303** | **0.231** | **0.323** | 0.233 | 0.325 | 0.225 | 0.312 | **0.224** | **0.310** |
| ETTh2 | ETTh1 | 0.413 | 0.419 | **0.412** | **0.419** | 0.469 | 0.470 | **0.418** | **0.432** | 0.487 | 0.465 | **0.422** | **0.425** |
| | ETTm1 | 0.762 | 0.540 | **0.742** | **0.534** | 0.987 | 0.618 | **0.842** | **0.573** | 0.814 | 0.574 | **0.706** | **0.533** |
| | ETTm2 | 0.209 | 0.302 | **0.208** | **0.299** | 0.248 | 0.327 | **0.236** | **0.322** | **0.216** | **0.308** | 0.222 | 0.308 |
| ETTm1 | ETTh1 | 0.495 | 0.474 | **0.493** | **0.472** | 0.541 | 0.497 | **0.527** | **0.488** | 0.530 | 0.484 | **0.502** | **0.468** |
| | ETTh2 | **0.331** | **0.376** | 0.333 | 0.377 | 0.347 | 0.395 | **0.324** | **0.378** | 0.320 | 0.372 | **0.316** | **0.368** |
| | ETTm2 | **0.179** | **0.262** | 0.180 | 0.262 | 0.190 | 0.276 | **0.185** | **0.271** | **0.195** | **0.276** | 0.197 | 0.277 |
| ETTm2 | ETTh1 | 0.511 | 0.485 | **0.469** | **0.460** | 0.544 | 0.508 | **0.500** | **0.477** | 0.549 | 0.497 | **0.471** | **0.453** |
| | ETTh2 | 0.306 | 0.364 | **0.297** | **0.351** | 0.298 | 0.362 | **0.294** | **0.354** | 0.311 | 0.369 | **0.294** | **0.352** |
| | ETTm1 | 0.411 | 0.408 | **0.359** | **0.380** | 0.445 | 0.429 | **0.373** | **0.390** | 0.411 | 0.416 | **0.357** | **0.378** |

