# OpenReview forum: "Patch-wise Structural Loss for Time Series Forecasting"
_ICML.cc/2025/Conference — ICML 2025 poster_

### Official Review · Reviewer_Pd9N · 2025-03-10

**Overall Recommendation:** 3

**Summary:**

Traditional loss functions, such as Mean Squared Error, often miss structural dependencies in time series forecasting. This paper proposes a Patch-wise Structural Loss to improve accuracy by focusing on patch-level structural alignment. It uses Fourier-based Adaptive Patching to divide the series and incorporates local statistical features—correlation, variance, and mean—with dynamic gradient weighting. Testing shows enhanced forecasting performance across multiple datasets and models.

**Claims And Evidence:**

There are no obvious problematic claims in the paper.

**Essential References Not Discussed:**

No essential reference not discussed.

**Experimental Designs Or Analyses:**

The design of the main experiments is comprehensive, as the PSloss is applied to various model architectures and achieves relatively good results. The ablation experiments are thorough, and their design and analysis help me better understand the method.

**Methods And Evaluation Criteria:**

The design and presentation of the method both make sense. The evaluation of the method also aligns with the established standards in the field.

**Other Comments Or Suggestions:**

No more comments or suggestions.

**Other Strengths And Weaknesses:**

Strengths:
1. The paper is well written and easy to understand.
2. The paper includes comprehensive evaluations. The diverse ablation study helps to understand the proposed approach.

Weaknesses:
1. The shortcomings of MSE presented in the paper are not only its limitations as an optimization objective but also as a metric. However, to align with the work in the field, the paper still uses MSE and MAE as main metrics. Figure 3 demonstrates how PSloss contributes to the final prediction results, but I believe the authors could provide more quantitative metrics to further illustrate this point.

**Questions For Authors:**

Question 1: Why use adaptive patching? What are the drawbacks of fixed-length patching?

**Relation To Broader Scientific Literature:**

1. Most current time series works, such as PatchTST[1] and iTransformer[2], use MSE loss as the optimization objective. This paper, however, highlights the limitations of using MSE loss for optimization.
2. The paper employs the Pearson Correlation Coefficient[3] to characterize correlation loss and the Kullback–Leibler (KL) divergence to characterize variance loss.
3. Inspired by previous works on balancing multi-task losses, such as [4-5], the paper proposes Gradient-based Dynamic Weighting to achieve balanced optimization.

References:
- [1] A Time Series is Worth 64 Words: Long-term Forecasting with Transformers
- [2]  iTransformer: Inverted Transformers Are Effective for Time Series Forecasting
- [3] Pearson correlation coefficient
- [4] Multi-task learning as multiobjective optimization.
- [5] Sparsetsf: Modeling long-term time series forecasting with* 1k* parameters.

**Theoretical Claims:**

No question about theoretical claims.

---

> ### Author Rebuttal · Authors · 2025-04-01
>
> Thank you very much for your valuable feedback. Below are our responses to your concerns and suggestions.
> # [W1] Additional quantitative metrics for evaluating PS loss performance
> To provide a more comprehensive evaluation, we incorporated additional metrics: **Dynamic Time Warping (DTW)**, **Time Distortion Index (TDI)**, and **Pearson Correlation Coefficient (PCC)**, to assess the performance of PS loss. A detailed explanation of these metrics is provided below:
>
> | Metric | Definition | Interpretation |
> |--|--|--|
> | Dynamic Time Warping (DTW) | Measures the minimum cumulative distance between two sequences after applying an optimal non-linear alignment. | Lower DTW indicates that the prediction closely matches the ground truth after optimal alignment. |
> | Time Distortion Index (TDI) | Quantifies the amount of temporal warping or distortion required to achieve the optimal alignment obtained by DTW. | Lower TDI indicates less temporal adjustments for optimal alignment, while a higher TDI signifies greater distortion.|
> | Pearson Correlation Coefficient (PCC) | Measures the linear relationship between two sequences. | Higher PCC indicates better preservation of the sequence's overall trend and structure. |
>
> On the iTransformer model, **PS loss consistently improves all three shape-aware metrics**, indicating better structural alignment. On the ETTh2 dataset, the iTransformer trained with MSE achieves lower DTW scores, reflecting smaller numerical differences after optimal alignment. However, the higher TDI in this case suggests that the alignment requires more extensive temporal warping, which indicates greater structural distortion compared to the forecasts generated using PS loss.
>
> |Metrics|DTW||TDI||PCC||
> |-|-|-|-|-|-|-
> |Loss|MSE|**+PS**|MSE|**+PS**|MSE|**+PS**
> |ETTh1|7.355|**7.324**|7.888|**6.959**|0.514|**0.530**
> |ETTh2|**6.891**|7.016|24.723|**22.705**|0.299|**0.342**
> |ETTm1|6.568|**6.435**|12.451|**11.381**|0.538|**0.557**
> |ETTm2|5.913|**5.611**|26.969|**22.495**|0.325|**0.387**
> |Weather|5.410|**5.409**|41.440|**40.343**|0.324|**0.352**
>
> Due to space limitations, we report the average metric values across all forecasting lengths. **Please refer to ([Table 6](https://anonymous.4open.science/r/PS_Re/T6.pdf)) for full results.**
>
> # [Q1] Drawbacks of fixed-length patching
>
> Fixed-length patching requires grid search over a predefined set of patch lengths, which introduces **computational overhead and lacks adaptability across datasets**.
>
> |Fixed Patch length|3|6|12|24|48|96|
> |-|-|-|-|-|-|-|
> |96|0.379|**0.378**|0.379|0.383|0.383|0.386|
> |192|0.430|0.429|**0.428**|0.431|0.431|0.433|
> |336|0.473|**0.473**|0.474|0.480|0.480|0.483|
> |720|0.496|0.499|**0.480**|0.493|0.493|0.508|
> |Avg.|0.444|0.445|**0.440**|0.446|0.450|0.453|
>
> In our ETTh1 experiments, the best-performing fixed patch size was found to be $P = 12$, which matches the patch length estimated by our adaptive patching strategy using the dominant period $p$. This demonstrates that our method can **automatically identify an appropriate patch length** without manual tuning.

---

### Official Review · Reviewer_jewV · 2025-03-11

**Overall Recommendation:** 4

**Summary:**

This paper proposes the Patch-wise Structural (PS) loss function for time series forecasting. The PS loss improves the alignment of local statistical properties (correlation, variance, and mean), addressing the limitations of traditional point-wise loss functions like MSE. By incorporating patch-level analysis, PS loss enhances the ability to model complex temporal structures. Extensive experiments on 7 real-world time series datasets demonstrate that PS loss significantly outperforms traditional methods, improving forecasting accuracy across various models, including LLM-based forecasting models.

## update after rebuttal
I support for accept

**Claims And Evidence:**

The claims made in the paper are well-supported by experimental evidence. The experiments in the submission and appendix demonstrate that the proposed method is effective and robust across different models and architectures. The source code and detailed experimental procedures further enhance the study's reproducibility.

**Essential References Not Discussed:**

The authors have effectively cited and discussed relevant work in time series forecasting, loss functions, and the patching mechanism in time series forecasting in the paper.

**Experimental Designs Or Analyses:**

The experimental design is solid and thorough, and the results support the claims made in the paper. However, there are several issues that need to be discussed:
1.  The paper does not provide a detailed comparison between the GDW strategy and grid search for loss coefficient selection. It would be helpful if the authors could offer a deeper analysis of the GDW strategy’s effectiveness.
2. The authors should provide visualizations or analyses showing how the different loss weights evolve during training, to better understand how the GDW strategy influences model optimization.

**Methods And Evaluation Criteria:**

The methods and evaluation criteria used in this paper are well-suited for the problem of time series forecasting. The authors choose relevant benchmark datasets and employ suitable evaluation metrics to assess model performance. The integration of PS loss with MSE is clearly explained, and the experimental setup is sound. The use of both quantitative and qualitative results strengthens the validity of the conclusions.

**Other Comments Or Suggestions:**

Minor inconsistency in notation: In Figure 2, \alpha is used for L_Corr, \beta for L_Mean, and \gamma for L_Var, but these notations are inconsistent with the rest of the manuscript and formula. It would be beneficial to unify the notation.

**Other Strengths And Weaknesses:**

Strengths:
1. The paper presents a clear and novel contribution to time series forecasting by addressing the limitations of traditional loss functions through the integrate seamless with PS loss, which provides a more precise method for structural alignment and achieves more practical predictions.
2. The gradient-based dynamic weighting strategy is a novel contribution that enhances the effectiveness of PS loss by adjusting the weight of each component based on gradient magnitudes, improving robustness without the computational cost of grid search.

Weaknesses:
1. The paper includes a sensitivity analysis of the hyperparameters \lamba and \delta, but it lacks a detailed explanation of how they vary in different scenarios. It is recommended that the authors clarify the parameter settings.

**Questions For Authors:**

1. About the ablation study, Could the authors explore whether PS loss can replace MSE loss entirely while achieving similar results in terms of forecasting accuracy?
2. Since PS loss involves local analysis, how does it affect training time, especially for large-scale datasets such as ECL? Would training time significantly increase for large datasets?
3. In the zero-shot forecasting experiment, do longer forecasting horizons benefit more from PS loss, or is its impact more significant in short-term forecasting?
4. How does PS loss compare with other loss functions such as FreDF (Learning to Forecast in Frequency Domain)? Does the combination of PS loss with these other functions enhance forecasting performance?

**Relation To Broader Scientific Literature:**

The proposed PS loss builds on existing time series forecasting loss functions but introduces a more flexible and localized approach. The introduction of patch-wise structural alignment is a novel contribution that distinguishes this work from existing methods. This innovation positions PS loss as a valuable contribution to the field of time series forecasting and makes it relevant to both academic research and practical applications.

**Theoretical Claims:**

I have checked the proofs and the theoretical claims behind PS loss, which are robust, and do not find any issues.

---

> ### Author Rebuttal · Authors · 2025-04-01
>
> Thank you very much for your valuable feedback. Below are our responses to your concerns and suggestions.
>
> # [E1] Comparison between GDW and grid-search
> To evaluate GDW against a traditional grid search for selecting loss weights, we conducted experiments on the ETTh1 dataset using iTransformer. Both methods use the same overall PS loss weight $\lambda$. For grid search, coefficients $\alpha$, $\beta$, and $\gamma$ were chosen from {0.3, 0.5, 0.7, 1.0}, totaling **64 runs per prediction length**. We report the best and average performance from grid search for comparison:
>
> |Method|GDW||Grid Search (Best)||Grid Search (Average)||
> |-|-|-|-|-|-|-|
> |Metric|MSE|MAE|MSE|MAE|MSE|MAE|
> |96|**0.379**|**0.396**|0.380|**0.396**|0.385|0.398|
> |192|**0.428**|**0.424**|**0.428**|**0.424**|0.432|0.426|
> |336|0.474|0.453|**0.473**|**0.450**|0.483|0.456|
> |720|**0.480**|**0.478**|0.483|0.479|0.513|0.494|
> |Avg|**0.440**|0.438|0.441|**0.437**|0.453|0.444|
>
> **GDW achieves performance comparable to the best grid-searched results, while avoiding exhaustive tuning**. Moreover, it reflects the intuition that the weights of correlation, variance, and mean should evolve dynamically during training to maintain balanced attention across all three loss terms, which static coefficients cannot assure.
>
> # [E2] Visualization of loss weights generated by GDW
> We visualize the evolution of weights generated by GDW in [Figure 1](https://anonymous.4open.science/r/PS_Re/F1.pdf) and made the following observations:
>
> - **Weight Range:** The weights for correlation, variance, and mean have different ranges, reflecting the inherent variation in their gradient magnitudes, which highlights the need for adaptive balancing.
> - **Weight Evolution:** Correlation weight tends to decrease, while variance and mean weights increase. This does not imply shifting focus, but rather ensures equilibrium among components, allowing structural alignment to be preserved during optimization.
>
> This confirms that GDW adaptively balances multiple objectives throughout training, improving stability and convergence.
>
> # [W1] Hyperparameter settings
> The PS loss weight $\lambda$ is selected from {0.1, 0.3, 0.5, 0.7, 1.0, 2.0, 3.0, 5.0, 10.0}. The patch length threshold $\delta$ is chosen from {24, 48}.
>
> # [Q1] PS loss as a standalone objective
> Results show that PS loss alone yields comparable accuracy to MSE+PS, demonstrating its effectiveness as a **standalone optimization objective** ([Table 8](https://anonymous.4open.science/r/PS_Re/T8.pdf)).
> |Model|iTransformer||TimeMixer||
> |-|-|-|-|-|
> |Loss|MSE + PS|PS Only|MSE+PS|PS Only|
> |ETTh1|0.440|**0.439**|0.437|**0.429**
> |ETTh2|**0.375**|0.380|0.369|**0.364**
> |ETTm1|**0.396**|**0.396**|**0.375**|0.377
> |ETTm2|**0.281**|0.282|**0.270**|0.274
> |Weather|**0.253**|**0.253**|0.243|**0.242**
>
> # [Q2] PS loss complexity on large datasets
> We report the empirical runtime cost by measuring the average **seconds per epoch** during training using **iTransformer** across three datasets: **ETTh1 (small), Weather (medium), and ECL (large)**:
> |Dataset|MSE|PS|Time Increase
> |-|-|-|-
> |ETTh1|1.96|2.66|0.71
> |Weather|10.63|14.04|3.40
> |ECL|25.02|30.20|5.18
>
> Despite the added cost, **the runtime increase is modest and justified by performance gains**.
>
> # [Q3] Zero-shot performance across forecasting lengths
>
> We extended zero-shot experiments based on iTransformer to forecast lengths of **96, 336, and 720**, in addition to 192 (reported in the paper). PS loss improved accuracy in **33 out of 36** settings, confirming its robustness across both short- and long-term horizons. See [Table 9](https://anonymous.4open.science/r/PS_Re/T9.pdf) for details.
>
> |Model|96||336||720||
> |-|-|-|-|-|-|-
> |Loss Function|MSE|+PS|MSE|+PS|MSE|+PS
> |ETTh1→ETTh2/m1/m2|0.499|**0.446**|0.537|**0.575**|0.579|**0.566**
> |ETTh2→ETTh1/m1/m2|0.626|**0.540**|0.641|**0.614**|0.686|**0.645**
> |ETTm1→ETTh1/h2/m2|0.420|**0.385**|0.514|**0.472**|0.550|**0.506**
> |ETTm2→ETTh1/h2/m1|0.622|**0.485**|0.792|**0.541**|0.870|**0.554**
> |Imp|-|**15.58%**|-|**11.39%**|-|**15.42%**
>
> # [Q4] Combination of PS loss and FreDF loss
> FreDF focuses on frequency-domain alignment to **mitigate label autocorrelation**, while PS loss emphasizes **patch-wise structural alignment** in the time domain. Their goals are complementary. We evaluated MSE+FreDF, MSE+PS, and MSE+PS+FreDF using iTransformer as backbone.
>
> |Loss|MSE+FreDF||MSE+PS||MSE+PS+FreDF||
> |-|-|-|-|-|-|-
> |Dataset|MSE|MAE|MSE|MAE|MSE|MAE
> |ETTh1|0.443|0.437|0.440|0.438|**0.436**|**0.433**
> |ETTm1|0.404|0.406|**0.396**|**0.397**|**0.396**|**0.397**
>
> Results show that combining the two losses yields **either improved or comparable performance**, supporting their compatibility. [Table 10](https://anonymous.4open.science/r/PS_Re/T10.pdf) provides full results.

---

> > ### Comment · Reviewer_jewV · 2025-04-03
> >
> > The authors have provided clear and satisfactory responses to my concerns. Their clarification of the generalization issue and the motivation behind modeling patch-level alignment is convincing. The novelty of the proposed framework is better justified, and the new experiments and metrics further support the method's effectiveness. I am now more confident in the contribution of this work and support its acceptance.

---

> > > ### Author Response · Authors · 2025-04-06
> > >
> > > Thank you very much for your thoughtful review of our work. We sincerely appreciate your valuable feedback and your confidence in our contribution. Thank you once again for your insightful suggestions and continued support!

---

### Official Review · Reviewer_YfmW · 2025-03-12

**Overall Recommendation:** 3

**Summary:**

Most previous time series forecasting models use MSE as the loss function, which treats each time step independently and neglect the structural dependency among steps. To fill the gap, this work proposes Patch-wise Structural (PS) Loss. PS Loss first splits target series into patches with patch size determined by FFT. Then correlation, variance and mean losses are computed within each patch and averaged. A gradient-based dynamic weighting mechanism is used to balance the weights of three losses during training. Experiments on real-world datasets show that PS loss can boost forecasting accuracy of both traditional and LLM-based models.

**Claims And Evidence:**

Claims are well supported.

**Essential References Not Discussed:**

Not applicable, references are generally comprehensive.

**Experimental Designs Or Analyses:**

1. Considering that FFT brings additional computation cost, how is the efficiency of PS loss w.r.t different forecasting lengths and channel numbers?
2. Can PS loss along without MSE be used as the loss function? How does it perform?

**Methods And Evaluation Criteria:**

Overall, the methods and evaluation are convincing. I have a few minor questions or concerns as follows:

1. PS loss utilizes FFT to detect the period in the target series for patching. What if there is no periodic pattern in the target? How does PS loss perform on such datasets (considering most datasets used in experiments have obviously daily periods)?
2. In gradient-based dynamic weighting, why does mean loss require further adjustment by Equation 12 among three losses? What is the purpose behind this design, and has there been an ablation study conducted on it?
3. PS loss focuses on structural dependency, but the evaluation metrics are still point-wise MSE and MAE, which may not effectively measure structural consistency. The authors should not be criticized for this, as they are standard metrics. However, I would still like to inquire whether there are other candidate metrics (like DTW) that could better reflect structural information.

**Other Comments Or Suggestions:**

See my comments and suggestions above.

**Other Strengths And Weaknesses:**

Overall, this is a commendable work. The topic is significant, as most recent studies have primarily focused on backbone designs, leaving loss functions relatively underexplored. The presentation is clear and easy to follow. Should the authors adequately address my concerns, I would be happy to raise the score.

**Questions For Authors:**

See my questions above.

**Relation To Broader Scientific Literature:**

The primary objective of this paper is to enhance time series forecasting, with outcomes that can be effectively applied across diverse downstream domains

**Theoretical Claims:**

Not applicable, no new theoretical claims is proposed.

---

> ### Author Rebuttal · Authors · 2025-04-01
>
> Thank you very much for your valuable feedback. Below are our responses to your concerns and suggestions.
> # [M1] PS loss performance on non-periodic targets
> When there is no clear periodic pattern in the target series, the dominant frequency—i.e., the one with the highest amplitude in the FFT spectrum—does not necessarily correspond to a true periodic component in the data. Instead, it typically falls into one of two categories:
> - **Short period ($p<\delta$)**: This often results from high-frequency components such as local fluctuations or noise. In this case, Equation (3) yields a short patch length, which still allows the model to focus on finer-grained local structure.
> - **Long period ($p>\delta$)**: This typically corresponds to low-frequency background components or weak global trends. In this case, the patch length will be capped at $\delta$  to prevent excessively large patches that could hinder fine-grained comparisons.
>
> This design allows PS loss to adapt to both periodic and non-periodic series, using frequency content to guide patch granularity, while $\delta$ prevents overly large patches. On the Exchange dataset, which lacks a clear periodic pattern, PS loss still improved MSE by **6.43%** on DLinear, demonstrating its effectiveness.
> # [M2] Purpose of mean loss refinement
> The purpose behind this design is to **first focus on aligning the shape of the series, and then gradually increase attention to the value offset**. As correlation and variance alignment improve during training (indicated by increasing $c$, $v$), the model allocates more weight to $L_{mean}$ to refine value-level offsets. Ablation on iTransformer and TimeMixer (ETTh1) confirms its effectiveness.
> |Method|iTrans*+PS||W/o c&v||TimeM*+PS||W/o c&v||
> |-|-|-|-|-|-|-|-|-
> |Metric|MSE|MAE|MSE|MAE|MSE|MAE|MSE|MAE
> |96|**0.379**|**0.396**|0.380|0.396|**0.366**|**0.392**|0.368|0.391
> |192|**0.428**|**0.424**|0.429|0.425|0.421|0.421|**0.418**|**0.420**
> |336|**0.474**|**0.453**|0.480|0.458|**0.489**|**0.453**|0.498|0.457
> |720|**0.480**|**0.478**|0.505|0.492|**0.474**|**0.463**|0.480|0.464
> |Avg|**0.440**|**0.438**|0.448|0.443|**0.438**|**0.432**|0.441|0.433
> # [M3] Additional metrics for structural evaluation
> Beyond MSE/MAE, we include additional shape-aware metrics: **DTW, TDI, and PCC**, using iTransformer for evaluation (Please see Reviewer Pd9N [W1] for metric details). **PS loss consistently improves all three metrics**, indicating better structural alignment ([Table 6](https://anonymous.4open.science/r/PS_Re/T6.pdf)).
> |Metrics|DTW||TDI||PCC||
> |-|-|-|-|-|-|-
> |Loss|MSE|+PS|MSE|+PS|MSE|+PS
> |ETTh1|7.355|**7.324**|7.888|**6.959**|0.514|**0.530**
> |ETTh2|**6.891**|7.016|24.723|**22.705**|0.299|**0.342**
> |ETTm1|6.568|**6.435**|12.451|**11.381**|0.538|**0.557**
> |ETTm2|5.913|**5.611**|26.969|**22.495**|0.325|**0.387**
> |Weather|5.410|**5.409**|41.440|**40.343**|0.324|**0.352**
> # [E4] Time complexity analysis of PS loss
> ### **1. Theoretical time complexity analysis**
> We analyze the time complexity of PS loss with respect to the forecast length $T$, number of channels $C$, and hidden dimension $d$. The overall complexity arises from three main components:
> - **Fourier-based Adaptive Patching (FAP):** The complexity of this component is dominated by the Fast Fourier Transform (FFT), which is $O(T\log T)$ per channel. Since FFT is applied to each of the $C$ channels, the total time complexity is $O(C\cdot T\log T)$.
> - **Patch-wise Structural Loss (PS):** The series is split into $N \approx \frac{2T}{P}$ patches, where $P$ is the patch length. Calculating correlation, variance, and mean over each patch requires $O(P)$ operations. Given $C\cdot N$ patches, the total complexity becomes $O(C\cdot N\cdot P) = O(C\cdot T)$.
> - **Gradient-based Dynamic Weighting (GDW):** The gradient computation for each loss component w.r.t. the model output has shape $d\cdot T$, leading to a complexity of $O(d\cdot T)$.
>
> Therefore, the overall time complexity of PS loss is **$O(C\cdot T\log T+C\cdot T+d\cdot T)$**.
> ### **2. Actual run time overhead**
> We report the empirical runtime cost by measuring **seconds per epoch** using **iTransformer** across three datasets: **ETTh1 (small), Weather (medium), and ECL (large)**:
> |Dataset|MSE|PS|Time Increase
> |-|-|-|-
> |ETTh1|1.96|2.66|0.71
> |Weather|10.63|14.04|3.40
> |ECL|25.02|30.20|5.18
>
> Despite the added cost, **the runtime increase is modest and justified by performance gains**.
> # [E5] PS loss performance without MSE
> Results show that PS loss alone yields comparable accuracy to MSE+PS, demonstrating its effectiveness as a **standalone optimization objective** ([Table 8](https://anonymous.4open.science/r/PS_Re/T8.pdf)).
> |Model|iTransformer||TimeMixer||
> |-|-|-|-|-|
> |Loss|MSE+PS|PS Only|MSE+PS|PS Only|
> |ETTh1|0.440|**0.439**|0.437|**0.429**
> |ETTh2|**0.375**|0.380|0.369|**0.364**
> |ETTm1|**0.396**|**0.396**|**0.375**|0.377
> |ETTm2|**0.281**|0.282|**0.270**|0.274
> |Weather|**0.253**|**0.253**|0.243|**0.242**

---

> > ### Comment · Reviewer_YfmW · 2025-04-04
> >
> > Thank you for your response, especially for conducting additional experiments during the rebuttal process. I will maintain my score of 3 and vote for acceptance. Moreover, I suggest that the experiments on complexity and new metrics (including their definitions, calculation methods, etc.), as well as the corresponding analysis, should be added to the final camera-ready version.

---

> > > ### Author Response · Authors · 2025-04-06
> > >
> > > Thank you very much for your thoughtful review of our work. We sincerely appreciate your valuable feedback and will ensure the additional experiments and analysis are incorporated into the final version. Thank you once again for your insightful suggestions and support!

---

### Official Review · Reviewer_K3Dy · 2025-03-12

**Overall Recommendation:** 3

**Summary:**

The authors propose a novel patch-wise structural (ps) loss, which is designed to enhance structural alignment by comparing time series at the patch level. By leveraging local statistical properties, e.g., correlation, variance, and mean, PS loss captures nuanced structural discrepancies overlooked by traditional point-wise loss. Experiments demonstrate that the PS loss can improve the performance of the state-of-the-art models across diverse real-world datasets.

## update after rebuttal
I support for accept.

**Claims And Evidence:**

The effectiveness of different module designs has been validated by the experimental results.

**Essential References Not Discussed:**

Some important baselines  e.g., Ada-MSHyper [1], and FAN [3], need to be compared. Please see Questions 1 and 2 for details.

[1] Shang Z, Chen L, Wu B, et al. Ada-MSHyper: adaptive multi-scale hypergraph transformer for time series forecasting. NeurIPS, 2024.

[3] Ye W, Deng S, Zou Q, et al. Frequency Adaptive Normalization For Non-stationary Time Series Forecasting. NeurIPS, 2024.

**Experimental Designs Or Analyses:**

The manuscript has some weaknesses in the experiments. Please see Questions for details.

**Methods And Evaluation Criteria:**

The proposed methods and evaluation criteria are appropriate for time series forecasting.

**Other Comments Or Suggestions:**

No

**Other Strengths And Weaknesses:**

1.The paper presents a notable innovation by exploring the integration of patch-wise structural loss into time series forecasting, a direction scarcely addressed by existing methodologies.

2.The organization of this paper is clear and the paper is well written.

**Questions For Authors:**

1. As the proposed loss function is specifically designed for time series forecasting, the comparative experiments should not only focus on long-term time series forecasting but also encompass other experimental settings, e.g., short-term time series forecasting and ultra-long-term time series forecasting, as mentioned by existing methods [1, 2].

2. Since some latest methods [1, 3] also improve model performance by introducing constraints or loss functions, as a loss specifically designed for time series, the authors should elaborate on the differences between their proposed loss function and these existing designs in the related work Section. In addition, more comparative experiments should be conducted to validate the effectiveness of the proposed loss functions against these advanced loss functions.

3. In Section 4.6, the authors demonstrate that the PS loss can improve generalization to unseen datasets. Given that LLMs have also been proven to exhibit strong generalization capabilities under zero-shot settings [4, 5, 6], it is suggested to study whether the PS loss can further enhance the performance of LLMs under zero-shot settings. The authors should validate this through additional experiments.

4.The design of the PS loss appears to be somewhat complex. The authors are suggested to include time complexity analysis.

5. To provide an intuitive understanding of the performance improvements brought by the proposed loss, the authors should explicitly list the performance gains in terms of percentage improvements.
[1]Shang Z, Chen L, Wu B, et al. Ada-MSHyper: adaptive multi-scale hypergraph transformer for time series forecasting. NeurIPS, 2024.
[2]Jia Y, Lin Y, Hao X, et al. WITRAN: Water-wave information transmission and recurrent acceleration network for long-range time series forecasting. NeurIPS, 2023.
[3]Ye W, Deng S, Zou Q, et al. Frequency Adaptive Normalization For Non-stationary Time Series Forecasting. NeurIPS, 2024.
[4]Zhou T, Niu P, Sun L, et al. One fits all: Power general time series analysis by pretrained LM. NeurIPS, 2024.
[5]Liu Y, Qin G, Huang X, et al. Autotimes: Autoregressive time series forecasters via large language models. NeurIPS, 2024.
[6]Jin M, Wang S, Ma L, et al. Time-LLM: Time Series Forecasting by Reprogramming Large Language Models. ICLR, 2024.

**Relation To Broader Scientific Literature:**

The introduce of patch-wise structural loss is beneficial for time series forecasting.

**Theoretical Claims:**

I have checked the correctness of the proofs for the theoretical claims.

---

> ### Author Rebuttal · Authors · 2025-04-01
>
> Thank you very much for your valuable feedback. Below are our responses to your concerns and suggestions.
> # [Q1] PS loss on ultra-long-term and short-term forecasting
> We evaluated PS loss on **ultra-long-term (T = {1080, 1440, 1800, 2160})** and **short-term (T = {12, 24, 48})** forecasting tasks using iTransformer and DLinear. We report averaged MSE results. Please refer to [Table 1](https://anonymous.4open.science/r/PS_Re/T1.pdf) and [Table 2](https://anonymous.4open.science/r/PS_Re/T2.pdf) for full results.
>
> - **Ultra-long-term**: MSE reduced by **7.38% (iTransformer)** and **11.01% (DLinear)**.
>
> |Model|iTransformer||Dlinear||
> |-|-|-|-|-
> |Loss Function|MSE|+PS|MSE|+PS
> |ETTh1|0.753|**0.693**|0.696|**0.628**
> |ETTh2|0.545|**0.494**|1.241|**1.127**
> |ETTm1|0.577|**0.536**|0.487|**0.474**
> |ETTm2|0.480|**0.466**|0.557|**0.463**
> |Imp.|-|**7.38%**|-|**11.01%**
>
> - **Short-term**: MSE reduced by **3.43% (iTransformer)** and **1.60% (DLinear)**.
>
> |Model|iTransformer||Dlinear||
> |-|-|-|-|-
> |Loss Function|MSE|+PS|MSE|+PS
> |PEMS03|0.110|**0.107**|0.239|**0.235**
> |PEMS04|0.105|**0.101**|0.283|**0.279**
> |Imp.|-|**3.43%**|-|**1.60%**
>
> These results demonstrate the effectiveness of PS loss across both **short-term** and **ultra-long-term** forecasting tasks.
>
> # [Q2] Comparison with Ada-MSHyper and FAN
> The contributions of Ada-MSHyper and FAN differ from our PS loss, as they address distinct challenges:
> - **Ada-MSHyper** introduces a **hypergraph transformer** with a **graph constraint loss** to enhance multi-scale interaction modeling through hypergraph learning.
> - **FAN** proposes a frequency-based adaptive **normalization method** to address both trend and seasonal non-stationary patterns.
> - **PS (Ours)** presents a novel **loss function** that enhances structural alignment between predictions and ground truth via patch-wise statistical metrics.
>
> We also combined PS loss with both methods. Please refer to [Table 3](https://anonymous.4open.science/r/PS_Re/T3.pdf) and [Table 4](https://anonymous.4open.science/r/PS_Re/T4.pdf) for the full results.
> - **Ada-MSHyper + PS.** PS loss improves the average performance by **8.28% (MSE) and 4.68% (MAE)**.
>
> |Method|Ada-MSHyper||Ada-MSHyper+PS||
> |-|-|-|-|-
> |Dataset|MSE|MAE|MSE|MAE
> |ETTh1|0.137|0.262|**0.132**|**0.254**
> |ETTh2|0.107|0.231|**0.105**|**0.227**
> |Imp.|-|-|**8.28%**|**4.68%**
>
> - **FAN + PS.** When using DLinear as the backbone, PS loss further improves the average performance of FAN by **2.07% (MSE) and 2.31% (MAE)**.
>
> |Method|Dlinear+FAN||Dlinear+FAN+PS||
> |-|-|-|-|-|
> |Dataset|MSE|MAE|MSE|MAE|
> |ETTh1|0.444|0.485|**0.439**|**0.479**
> |ETTh2|0.137|0.262|**0.132**|**0.254**
> |Imp.|-|-|**2.07%**|**2.31%**
>
> These results demonstrate that while **FAN and Ada-MSHyper focus on different aspects, PS loss can still further improve their performance** by enhancing the structural alignment of the forecasted series.
>
> # [Q3] PS loss on LLM-based models for zero-shot forecasting
> We conducted zero-shot forecasting experiments with LLM-based models: OFA, AutoTimes, and Time-LLM. PS loss improved forecasting accuracy with average MSE reductions of **2.07% (OFA)**, **6.33% (AutoTimes)**, and **7.29% (Time-LLM)**. Please refer to [Table 5](https://anonymous.4open.science/r/PS_Re/T5.pdf) for the full results.
> |Model|OFA||AutoTimes||Time-LLM||
> |-|-|-|-|-|-|-
> |Loss Function|+MSE|+PS|+MSE|+PS|+MSE|+PS
> |ETTh1→ETTh2/m1/m2|0.410|**0.417**|0.421|**0.421**|0.420|**0.405**
> |ETTh2→ETTh1/m1/m2|0.461|**0.454**|0.568|**0.499**|0.506|**0.450**
> |ETTm1→ETTh1/h2/m2|0.335|**0.336**|0.359|**0.346**|0.349|**0.338**
> |ETTm2→ETTh1/h2/m1|0.411|**0.359**|0.445|**0.373**|0.424|**0.374**
> |Imp|-|**2.07%**|-|**6.33%**|-|**7.29%**
>
>
>
> # [Q4] Time complexity analysis of PS loss
> We analyze the time complexity of PS loss with respect to the forecast length $T$, number of channels $C$, and hidden dimension $d$. The overall complexity arises from three main components:
> - **Fourier-based Adaptive Patching (FAP):** The complexity of this component is dominated by the Fast Fourier Transform (FFT), which is $O(T\log T)$ per channel. Since FFT is applied to each of the $C$ channels, the total time complexity is $O(C\cdot T \log T)$.
> - **Patch-wise Structural Loss (PS):** The series is split into $N \approx \frac{2T}{P}$ patches, where $P$ is the patch length. Calculating correlation, variance, and mean over each patch requires $O(P)$ operations. Given $C \cdot N$ patches, the total complexity becomes $O(C \cdot N \cdot P) = O(C\cdot T)$.
> - **Gradient-based Dynamic Weighting (GDW):** The gradient computation for each loss component w.r.t. the model output has shape $d \cdot T$, leading to a complexity of $O(d \cdot T)$.
>
> Therefore, the overall time complexity of PS loss is **$O(C \cdot T \log T + C \cdot T + d \cdot T)$**.
> # [Q5] Performance gains in terms of percentage improvements
> We now report percentage improvements throughout the paper and summarize them in [updated Table](https://anonymous.4open.science/r/PS_Re/T7.pdf).

---

> > ### Comment · Reviewer_K3Dy · 2025-04-05
> >
> > Thank you for your response, especially for conducting additional experiments during the rebuttal process. I will maintain my score of 3 and vote for acceptance.

---

> > > ### Author Response · Authors · 2025-04-06
> > >
> > > Thank you very much for your thoughtful review of our work. We are sincerely grateful for your valuable feedback and your recognition of the additional experiments. Thank you once again for your insightful suggestions and support!

---

### Decision · Program_Chairs · 2025-05-01

**Decision:**

Accept (poster)

**Comment:**

All reviewers support acceptance, highlighting significant methodological novelty, comprehensive validation, and clear improvements, while recommending minor revisions and clarifications for final submission.